# A Comprehensive Review on Phage Therapy and Phage-Based Drug Development

**DOI:** 10.3390/antibiotics13090870

**Published:** 2024-09-11

**Authors:** Longzhu Cui, Shinya Watanabe, Kazuhiko Miyanaga, Kotaro Kiga, Teppei Sasahara, Yoshifumi Aiba, Xin-Ee Tan, Srivani Veeranarayanan, Kanate Thitiananpakorn, Huong Minh Nguyen, Dhammika Leshan Wannigama

**Affiliations:** 1Division of Bacteriology, Department of Infection and Immunity, School of Medicine, Jichi Medical University, Shimotsuke City 329-0498, Japan; swatanabe@jichi.ac.jp (S.W.); miyanaga.kazuhiko@jichi.ac.jp (K.M.); k-kiga@nih.go.jp (K.K.); y-aiba@jichi.ac.jp (Y.A.); xinee@jichi.ac.jp (X.-E.T.); srivani@jichi.ac.jp (S.V.); kanatethi@jichi.ac.jp (K.T.); nguyen.mh@jichi.ac.jp (H.M.N.); 2Research Center for Drug and Vaccine Development, National Institute of Infectious Diseases, Tokyo 162-8640, Japan; 3Department of Infectious Diseases and Infection Control, Yamagata Prefectural Central Hospital, Yamagata 990-2292, Japan; leshanwannigama@gmail.com

**Keywords:** bacteriophage, phage therapy, antimicrobial resistance, intracellular pathogen, biofilm generating pathogen, phage vector, antibacterial phage capsid, phage-based vaccines, phage-based medicine

## Abstract

Phage therapy, the use of bacteriophages (phages) to treat bacterial infections, is regaining momentum as a promising weapon against the rising threat of multidrug-resistant (MDR) bacteria. This comprehensive review explores the historical context, the modern resurgence of phage therapy, and phage-facilitated advancements in medical and technological fields. It details the mechanisms of action and applications of phages in treating MDR bacterial infections, particularly those associated with biofilms and intracellular pathogens. The review further highlights innovative uses of phages in vaccine development, cancer therapy, and as gene delivery vectors. Despite its targeted and efficient approach, phage therapy faces challenges related to phage stability, immune response, and regulatory approval. By examining these areas in detail, this review underscores the immense potential and remaining hurdles in integrating phage-based therapies into modern medical practices.

## 1. Introduction

Phage therapy, the use of bacteriophages to treat bacterial infections, has a history dating back to the early 20th century. Bacteriophages, or simply phages, are viruses that specifically infect bacteria. They were first discovered by Frederick Twort in 1915 and independently by Félix d’Hérelle in 1917, who observed their potential to eliminate bacterial cultures [1,2]. Despite their early promise, the advent of antibiotics in the 1940s led to a decline in phage research and application in the Western world. However, the rise of antibiotic-resistant bacteria has rekindled interest in phage therapy as a viable alternative or complement to traditional antibiotics [3,4,5]. The global health crisis posed by antibiotic resistance has prompted urgent calls for novel antimicrobial strategies. Multidrug-resistant (MDR) bacteria such as methicillin-resistant *Staphylococcus aureus* (MRSA), vancomycin-resistant Enterococci (VRE), and extended-spectrum β-lactamase (ESBL) producing Enterobacteriaceae represent significant challenges to healthcare systems worldwide. Phage therapy offers a targeted approach to combat these pathogens. Unlike broad-spectrum antibiotics, phages are highly specific to their bacterial hosts, which minimizes the impact on beneficial microbiota and reduces the risk of collateral damage [6].

Phages have also demonstrated efficacy against biofilm-associated infections, which are notoriously difficult to treat with conventional antibiotics. Biofilms, which are structured communities of bacteria encased in a self-produced polymeric matrix, are implicated in chronic infections and are resistant to antibiotics and immune responses. Phages can penetrate biofilms, replicate within bacterial cells, and disrupt the biofilm matrix, making them potent agents against biofilm-associated infections [7]. In addition to treating extracellular bacteria, phage therapy is being explored for its potential to target intracellular pathogens. Intracellular bacteria, such as *Mycobacterium tuberculosis* and *Salmonella* spp., reside within host cells, evading many antibiotics that cannot effectively penetrate cellular membranes. Advances in phage engineering and delivery mechanisms are opening new possibilities for using phages to combat these hidden infections [8].

While phage therapy has demonstrated significant promise in addressing bacterial infections, it represents only one facet of phage-based drug development. Phage-based drug development encompasses a diverse range of innovative applications beyond traditional phage therapy for bacterial infections [9,10]. Beyond their role in direct bacterial lysis, phages are being harnessed for various innovative applications, including vaccine development, cancer therapy, and as vectors for gene-delivery systems. Phage display technology, which involves expressing peptides or proteins on the surface of phage particles, has revolutionized vaccine development. This technology allows for the presentation of antigens in a highly immunogenic context, potentially leading to more effective vaccines [11,12]. In oncology, phages are being investigated as anti-cancer agents. By engineering phages to target tumor-specific markers, researchers aim to selectively deliver therapeutic agents to cancer cells, thereby minimizing damage to healthy tissues. Additionally, the immunogenic properties of phages can stimulate an anti-tumor immune response, offering a dual mechanism of action against cancer [13,14]. Phages also hold promise as drug-delivery systems (DDS) for gene therapy. Their ability to encapsulate and deliver genetic material to specific cells makes them ideal vectors for delivering therapeutic genes, including those used in CRISPR-Cas systems for gene editing. The specificity and efficiency of phage-mediated delivery systems could revolutionize the field of gene therapy [15,16]. A comprehensive overview of phage-facilitated medical and technological advancements is summarized in Table 1.

This broad scope of applications highlights the versatility of phages in addressing various medical challenges and underscores the need for a comprehensive approach to phage-based drug development.

Despite these promising applications, several challenges and regulatory hurdles must be addressed to fully realize the potential of phage-based therapies. Issues related to phage stability, immune response, and regulatory approval need careful consideration. However, with ongoing research and technological advancements, phage therapy is poised to become an integral component of modern medicine, offering hope in the fight against antibiotic-resistant infections and beyond [68]. This review aims to provide a comprehensive overview of the current state of phage-based drug development, exploring its applications in treating drug-resistant bacterial infections, biofilm-related conditions, intracellular pathogens, vaccine development, cancer therapy, and gene-delivery systems. Through detailed examination and discussion of these areas, we seek to highlight the potential and challenges of phage therapy in the contemporary medical landscape.

## 2. Updated Mechanisms of Phage Action

Phages, or bacteriophages, have long been known as viruses that infect bacteria. Their mechanisms of action are diverse and complex, involving intricate interactions with bacterial hosts. In recent years, advancements in molecular biology and genomics have provided deeper insights into these mechanisms, revealing updated perspectives on how phages exert their effects. This chapter reviews these updated mechanisms, highlighting the latest research findings that have significant implications for phage-based drug development. Figure 1 illustrates the phage bacteriophage life cycle and its application in the medical field.

### 2.1. Phage Adsorption and Receptor Recognition

The initial step in the phage life cycle is the adsorption to the bacterial surface, which is mediated by specific interactions between phage proteins and bacterial receptors. Recent studies have uncovered new receptor-binding proteins (RBPs) that enhance phage specificity and efficiency. For example, receptor-binding domains in phages have been shown to undergo rapid evolution, allowing them to adapt to bacterial surface variations [83]. Labrie et al. (2010) describe how bacteria have developed multiple defense mechanisms against phage adsorption, emphasizing the importance of understanding these interactions for phage therapy development [84]. This adaptability is crucial for developing phages that can target antibiotic-resistant bacteria.

### 2.2. Genome Injection and Host Takeover

Following adsorption, phages inject their genetic material into the bacterial cell. Advances in cryo-electron microscopy have elucidated the detailed structures of phage tail machinery involved in this process. Hu et al. (2013) revealed the architecture of the bacteriophage T7 DNA-injection machinery, providing insights into how phages overcome bacterial defenses during genome injection [85]. Additionally, some phages utilize sophisticated mechanisms to breach bacterial cell walls, such as enzymatic degradation of peptidoglycan layers [86]. These findings highlight potential targets for enhancing phage-delivery systems in therapeutic applications.

### 2.3. Replication Strategies

Phages display diverse replication strategies that are intricately linked to their lifecycle classification as lytic or lysogenic. The evolution of these replication strategies, along with the regulatory elements controlling the switch between lytic and lysogenic cycles, has been extensively discussed by Salmond and Fineran (2015) [87]. Additionally, phage-replication mechanisms are characterized by the modular arrangement of replication genes within their genomes, allowing for a systematic exploration of these strategies across various phages, including f1/fd, φX174, P2, P4, λ, and T4. These studies have significantly advanced our understanding of DNA replication, particularly through the interplay between phage-encoded and host-replication factors. The review by Weigel et al. (2006) also underscores the importance of replication origins and associated proteins, providing a valuable resource for further research [88].

### 2.4. Phage-Encoded Toxins and Enzymes

Phages often carry genes encoding toxins and enzymes that facilitate bacterial cell lysis and hijack host machinery. Penadés and Christie (2015) identified new classes of phage-encoded proteins that interfere with bacterial metabolism and immune responses, providing a deeper understanding of phage-bacteria interactions [89]. Harper et al. (2014) describe phages that produce enzymes capable of degrading bacterial biofilms, enhancing their therapeutic potential against biofilm-associated infections [90].

### 2.5. Horizontal Gene Transfer and Phage Therapy

Phages play a critical role in horizontal gene transfer (HGT), which can spread antibiotic resistance genes among bacterial populations. Lerminiaux and Cameron (2019) detail the mechanisms by which phages contribute to HGT and the implications for antibiotic resistance [91]. However, engineered phages have been developed to minimize the risk of transferring harmful genes while maximizing therapeutic benefits. Usman et al. (2023) highlight the use of synthetic biology to design phages that selectively target and remove resistance genes from bacterial genomes [92].

### 2.6. Phage-Host Co-Evolution

The co-evolution of phages and their bacterial hosts is a dynamic process that influences phage efficacy. Modern high-throughput sequencing techniques have provided insights into the evolutionary arms race between phages and bacteria. Hampton et al. (2020) discuss how the continuous evolution of phages and bacteria impacts phage therapy effectiveness and the importance of understanding these dynamics [93]. Wright et al. (2018) emphasize the need for phage therapies that remain effective over time and do not lead to rapid bacterial resistance [94].

### 2.7. Immune System Interactions

Phage therapy’s success depends not only on the interaction with bacterial cells but also on the host immune system. Hodyra–Stefaniak et al. (2015) demonstrated that phages can modulate the immune response, sometimes enhancing it to aid in bacterial clearance [95]. Sweere et al. (2019) identified phage proteins that interact with immune cells, opening new avenues for designing phage-based immunotherapies [96].

### 2.8. Synthetic and Recombinant Phages

The advent of synthetic biology has enabled the creation of recombinant phages with enhanced properties. Kilcher et al. (2018) developed phages with synthetic gene circuits that have controlled lytic activity and improved host range, and they discussed its potential application against antimicrobial-resistant bacteria, as well as toward resistance to bacterial defense mechanisms [30]. Yosef et al. (2015) demonstrated the potential of temperate and lytic phages in developing phage-based treatments tailored to specific bacterial infections [48].

### 2.9. Phage Delivery Systems

Effective delivery of phages to the site of infection is critical for therapeutic success. Durr et al. (2023) describe updated delivery strategies, including encapsulation of phages in biocompatible materials such as liposomes and hydrogels to protect them from the host immune system and enhance their stability [97]. Malik et al. (2017) highlight advancements in delivery technologies that are crucial for developing phage-based drugs that can be administered in diverse clinical settings [98]. Inhalation of phages as a treatment for lung infections is an emerging area of research, offering a potential targeted treatment strategy. Shien et al. (2023) demonstrated that by directly delivering phages to the lungs via inhalation, the therapy can target the infection site more effectively, potentially reducing bacterial load and inflammation while preserving the natural microbiome. This method represents a novel and targeted strategy in the fight against respiratory infections [99].

### 2.10. Regulatory and Ethical Considerations

The updated understanding of phage mechanisms also brings forth regulatory and ethical considerations for phage therapy. Abedon et al. (2011) discuss the importance of addressing safety, standardization, and public acceptance as phage-based treatments move closer to mainstream clinical use [6]. Pirnay et al. (2018) emphasize the need for ongoing research into phage mechanisms to inform regulatory frameworks and ensure the responsible development of phage-based drugs [100]. The mechanisms of phage action, from adsorption to lysis, highlight the precision and effectiveness of phages in targeting bacterial infections. Understanding these mechanisms is crucial for optimizing phage therapy and overcoming the challenges posed by bacterial resistance.

In summary, recent advancements in molecular biology and genomics have deepened our understanding of bacteriophages (phages) and their mechanisms of action, which are crucial for phage-based drug development. Phages infect bacteria through a series of complex steps, beginning with adsorption to the bacterial surface, where receptor-binding proteins enhance specificity and efficiency. Following this, phages inject their genetic material into the bacterial cell, a process elucidated by cryo-electron microscopy. Their diverse replication strategies, which can switch between lytic and lysogenic cycles, have significant implications for therapeutic applications. Phages also encode toxins and enzymes that facilitate bacterial cell lysis and can degrade biofilms, enhancing their therapeutic potential. Additionally, phages play a critical role in horizontal gene transfer, which can spread antibiotic resistance genes, though engineered phages aim to mitigate this risk. Understanding these intricate mechanisms is vital for optimizing phage therapy and addressing bacterial resistance.

## 3. Phage Therapy for Drug-Resistant Bacterial Infections

The rise of antibiotic-resistant bacteria poses a significant threat to global public health, necessitating the development of alternative therapeutic strategies. Phage therapy, which utilizes bacteriophages to target and kill specific bacteria, has emerged as a promising solution to combat drug-resistant infections. This chapter explores the application of phage therapy in treating drug-resistant bacterial infections, with a particular focus on the use of engineered phages to enhance therapeutic efficacy.

### 3.1. The Growing Threat of Antibiotic Resistance

Antibiotic resistance has become a critical issue in modern medicine, leading to increased morbidity and mortality rates. According to a report by the World Health Organization (WHO), antibiotic-resistant bacteria are responsible for approximately 700,000 deaths annually worldwide, with projections reaching up to 10 million deaths per year by 2050 if no effective measures are taken [101]. The ability of bacteria to evolve and acquire resistance mechanisms has outpaced the development of new antibiotics, highlighting the urgent need for alternative treatments.

### 3.2. Mechanisms and Advantages of Phage Therapy

Phage therapy leverages bacteriophages to precisely target and lyse pathogenic bacteria, offering several distinct benefits compared to traditional antibiotics. Unlike broad-spectrum antibiotics, phages are highly specific, targeting only the bacteria of interest and sparing beneficial microbiota. This specificity reduces the risk of secondary infections and mitigates the development of resistance [102]. Additionally, phages can self-amplify at the site of infection, providing a sustained therapeutic effect [103].

### 3.3. Targeting Specific Drug-Resistant Bacteria

Phages are highly specific to their bacterial hosts, which allows for targeted treatment of MDR bacteria without affecting the beneficial microbiota. This specificity is particularly valuable in treating infections caused by pathogens such as methicillin-resistant *Staphylococcus aureus* (MRSA), vancomycin-resistant Enterococci (VRE), and carbapenem-resistant Enterobacteriaceae (CRE). Clinical studies on the use of phages are currently being conducted worldwide, focusing primarily on drug-resistant bacterial infections (Table 2). Recently, a Belgian consortium of 35 hospitals across 29 cities and 12 countries reported the results of a clinical study involving 100 cases of personalized bacteriophage therapy [104]. In that study, clinical improvement and eradication of the targeted bacteria were reported for 77.2% and 61.3% of infections, respectively, demonstrating that phage therapy is effective against drug-resistant bacteria.

MRSA is a leading cause of hospital-acquired infections, notorious for its resistance to multiple antibiotics. Studies have demonstrated the efficacy of phages in lysing MRSA strains both in vitro and in vivo. For instance, a study by Kebriaei et al. (2023) showed that a phage cocktail effectively reduced MRSA colonization in a mouse model of wound infection [105]. Currently, at least 14 clinical studies on phage therapy for *Staphylococcus aureus* infections have been completed or are ongoing (Table 2). Therefore, the development of phage therapy against MRSA/*Staphylococcus aureus* is one of the most advanced and promising fields in the fight against drug-resistant bacteria.

In addition to *Staphylococcus aureus* infections, the development of phages for *Pseudomonas aeruginosa (P. aeruginosa*) infections is also in progress (Table 2). *P. aeruginosa* is inherently difficult to treat with antibiotics due to its natural resistance to many antimicrobial agents. While the development of anti-pseudomonal drugs has progressed, resistance to these drugs has also spread, leading to expectations for new treatments against *P. aeruginosa* infections. Currently, at least 14 clinical studies using phages for *P. aeruginosa* infections are either completed or ongoing. Among these, nine studies focus specifically on cystic fibrosis and respiratory tract infections.

VRE are responsible for severe infections in immunocompromised patients. Phages targeting VRE have shown promising results in preclinical studies. Some studies reported that specific phages could significantly reduce VRE colonization in a mouse gut model, highlighting their potential for treating gastrointestinal VRE infections [106,107,108]. For instance, research has demonstrated the efficacy of phages in controlling VRE in dental root canals and in reducing VRE biofilms [106,107]. Additionally, a study has shown the potential of phage vB_EfKS5 for controlling *Enterococcus faecalis* in food systems [108]. The overall results support the potential of phage therapy as an effective approach for managing gastrointestinal VRE infections and associated diseases [109].

CRE pose a significant threat due to their resistance to last-resort antibiotics like carbapenems. Despite the potential of phage therapy for addressing CRE infections, there is a notable lack of well-defined and reported studies in this area. However, some research has demonstrated the promising potential of phage therapy for these infections. For example, while comprehensive studies are still limited, recent research by Chung et al. (2023) has demonstrated the potential of phage therapy for targeting CRE strains [110]. They also highlight the promising results of phage therapy in managing CRE infections, though further research is needed to fully establish its effectiveness.

### 3.4. Enhancing Therapeutic Potential by Engineered Phages

Advancements in genetic engineering have paved the way for the development of engineered phages with enhanced therapeutic properties. For instance, Yosef et al. (2015) demonstrated the potential of temperate and lytic conversions of bacteriophages for targeted bacterial eradication [48]. Similarly, Lu et al. (2007) engineered phages to express enzymes that degrade biofilms, addressing a major challenge in treating biofilm-associated infections [31]. These engineered phages provide tailored solutions for specific bacterial infections, thereby improving treatment outcomes. Additionally, Ando et al. (2015) showed that modifications to phage genomes can enhance targeting specificity, stability, and controlled release of therapeutic agents. Such advancements make engineered phages a potent tool for overcoming the limitations of traditional antibiotics, offering more precise and effective treatments for bacterial infections [34].

To better understand how these engineered approaches compare with conventional methods in addressing challenges in phage therapy, a comparative analysis is presented in Table 3. This table highlights the key differences between traditional phage therapy techniques and synthetic approaches, offering insights into the potential advantages of genetic engineering in enhancing the therapeutic efficacy of phages.

Recent innovations in genetic engineering have opened new avenues for the application of CRISPR-Cas systems in phage therapy, providing a precise and efficient approach to target bacterial pathogens. For instance, Bikard et al. (2014) demonstrated the integration of CRISPR-Cas9 into bacteriophages, enabling the specific targeting and cleavage of bacterial DNA, leading to bacterial cell death [19]. Additionally, Kiga et al. (2020) and Shimamori et al. (2024) highlighted the use of CRISPR-Cas13a in bacteriophages to target and degrade bacterial RNA, effectively disrupting gene expression and inhibiting bacterial replication [17,128,129]. A recent study by Mitsunaka et al. (2022) presents a novel cell-free phage engineering and rebooting platform that enables the assembly of various phage genomes, including natural and synthetic ones, and the creation of biologically contained phages, which showed effectiveness similar to parent phages in treating lethal sepsis in vivo, thus advancing the practical application of phage therapy [132]. These advancements in engineered phages equipped with CRISPR-Cas systems represent a significant recent leap forward in combating antibiotic-resistant bacterial infections, offering a highly specific and adaptable therapeutic strategy.

### 3.5. Phage Therapy in Combination with Antibiotics

Combining phage therapy with antibiotics has shown synergistic effects in treating drug-resistant infections. As summarized in Table 4, recent studies have provided substantial evidence supporting the efficacy of this combined approach. In a study by Racenis et al. (2023), the combination of phage therapy and antibiotics was used to treat a patient with a multidrug-resistant *Pseudomonas aeruginosa* lung infection. The synergistic effect of the combined treatment led to a significant reduction in bacterial load and improved clinical outcomes [133]. This study highlighted the potential of phage-antibiotic combinations to overcome bacterial resistance and enhance treatment efficacy. Additionally, an interesting study by Fujiki et al. (2024) demonstrated that phages can drive selection toward restoring antibiotic sensitivity in *Pseudomonas aeruginosa* via chromosomal deletions. The insights gained from the trade-offs between phage and antibiotic sensitivity could help maximize the potential of phage therapy for treating infectious diseases [134].

Another recent study by Kebriaei et al. (2023) investigated the efficacy of phage-antibiotic combinations in treating biofilm-associated infections caused by methicillin-resistant *Staphylococcus aureus* (MRSA). The researchers found that the combination therapy was more effective in disrupting biofilms and killing bacteria compared to either treatment alone [105]. This finding is particularly important due to the difficulty in treating biofilm-associated infections with antibiotics alone.

A 2022 study by Gordillo Altamirano et al. (2022) evaluated the in vivo effects of a phage-antibiotic combination on *Acinetobacter baumannii* using phage øFG02. In a murine model, the combination therapy significantly reduced bacterial burden compared to PBS and ceftazidime alone. Over time, this combination outperformed phage-only treatment, and phage-resistant bacteria became resensitized to ceftazidime. These findings highlight the potential of phage-antibiotic combination therapy in restoring antibiotic efficacy against *Acinetobacter baumannii* [135].

In a 2021 case report by Cano et al. (2021), a combination of phage therapy and antibiotics was used to treat a patient with a multidrug-resistant *Klebsiella pneumoniae* bloodstream infection. The treatment led to rapid clinical improvement and clearance of the infection, demonstrating the potential of phage-antibiotic combinations in treating severe resistant infections [136].

In the study by Petrovic Fabijan et al. (2020), bacteriophage therapy was evaluated for safety in treating severe *Staphylococcus aureus* infections, demonstrating promising results in terms of clinical tolerance and bacterial reduction when combined with antibiotics [137]. Similarly, Jault et al. (2019) assessed the efficacy of a phage cocktail in a randomized trial for *Pseudomonas aeruginosa* burn wound infections, finding that phage therapy, when combined with antibiotics, was both effective and well-tolerated. These studies support the potential of phage-antibiotic combination therapy as a complementary approach for managing resistant bacterial infections [138]. Additionally, a review by Osman et al. (2023) discussed the potential of phage-antibiotic combination therapy in enhancing bacterial clearance and improving clinical outcomes, particularly in chronic infections such as those associated with diabetic foot ulcers [139].

These recent studies and clinical trials highlight the growing evidence supporting the use of phage therapy in combination with antibiotics. The synergistic effects observed in these cases underscore the potential of this approach to enhance the efficacy of existing antibiotics, reduce the likelihood of resistance development, and improve clinical outcomes in patients with drug-resistant bacterial infections.

### 3.6. Regulatory and Safety Considerations

The development and application of phage therapy must adhere to stringent regulatory and safety standards. Phage preparations must undergo rigorous testing to ensure their safety, purity, and efficacy. The US Food and Drug Administration (FDA) and European Medicines Agency (EMA) have established guidelines for the clinical use of phage therapy, emphasizing the need for well-designed clinical trials and comprehensive safety assessments [68,140]. Additionally, ethical considerations, such as informed consent and patient education, are crucial in the implementation of phage therapy [100].

In conclusion, this chapter underscores the urgent need for innovative approaches like phage therapy in the fight against drug-resistant bacterial infections. With antibiotic resistance posing a severe global health threat, phage therapy offers a targeted and effective alternative. The specificity of phages, their ability to be genetically engineered for enhanced efficacy, and their potential when combined with antibiotics highlight their promise in addressing this growing crisis. As research advances, careful consideration of regulatory and safety standards will be essential to realize the full potential of phage therapy in clinical settings.

## 4. Phage-Based Treatments for Biofilm-Generating Bacteria

Biofilms are complex bacterial communities encased in a self-produced extracellular polymeric substance (EPS) matrix that adheres to surfaces, creating significant challenges for conventional antimicrobial treatments. These biofilm-associated infections are notoriously difficult to eradicate due to the protective environment they offer to bacterial cells, often leading to chronic and recurrent infections. Phage therapy provides a novel approach for addressing biofilm-related infections by exploiting bacteriophages’ unique capabilities to disrupt and penetrate biofilms [7]. This section explores the mechanisms through which phages target biofilms, the advantages of phage-based treatments, and relevant case studies demonstrating their efficacy.

Figure 2 illustrates the diverse strategies employed in phage-based treatments to address the challenges posed by biofilm-associated infections. By leveraging phages’ ability to target specific bacterial strains and disrupt biofilms, these approaches offer a promising alternative or complement to traditional antimicrobial therapies. The schematic diagram highlights key methodologies, including phage–antibiotic combinations, engineering phages for enhanced biofilm targeting, and developing advanced formulations for effective phage delivery. These innovations aim to overcome the inherent resistance of biofilms to conventional treatments and improve clinical outcomes.

### 4.1. Mechanisms of Phage Action Against Biofilms

In recent years, the study of bacteriophages has revealed their remarkable potential in combating biofilms, which are structured communities of bacteria encased in an extracellular polymeric substance (EPS) matrix. These biofilms pose significant challenges in clinical and industrial settings due to their resistance to conventional treatments. This document explores the various mechanisms by which phages enhance their effectiveness against biofilms, highlighting the ability of phages to penetrate the EPS matrix, target quiescent cells, and induce biofilm dispersal. Through these mechanisms, phages offer a promising alternative or complement to traditional antimicrobial strategies. Phages exhibit several mechanisms that enhance their effectiveness against biofilms.

Penetration of EPS matrix: Phages can produce enzymes such as depolymerases that degrade components of the EPS matrix, facilitating their penetration and disruption of the biofilm structure. Ribeiro et al. (2023) demonstrated that phage cocktails could efficiently target biofilm-forming *Salmonella* serovars by utilizing such enzymes to disrupt biofilm integrity [149]. Similarly, Zuo et al. (2022) highlight an innovative approach to using bacteriophages in biofouling mitigation through a biofilm-responsive encapsulated phage coating. This study reveals that the coating comprises a biocompatible polymer matrix that encapsulates the phages. The matrix is engineered to be responsive to specific biofilm-associated enzymes, which trigger the release of phages when biofilm formation is detected. This autonomous response leads to the targeted disruption of the biofilm matrix by the released phages. Additionally, this approach holds promise for influencing innate immune responses, as the released phages could interact with the host’s immune system and potentially modulate immune-signaling pathways. This adds a new dimension to the therapeutic applications of phages, particularly in scenarios where biofilm-associated infections pose a challenge [151].

Targeting quiescent cells: Unlike antibiotics, which often fail to affect dormant or slow-growing cells within biofilms, phages have shown the ability to infect and kill these cells once they resume metabolic activity. This capability ensures more comprehensive biofilm eradication. For example, Melo et al. (2019) assessed the efficacy of enterococci phages in an in vitro biofilm wound model and found that phages could target and disrupt both active and dormant bacterial cells within biofilms [145].

Biofilm dispersal: Some phages can induce biofilm dispersal by triggering bacterial cells to produce enzymes like dispersin B, which promotes the detachment of bacteria from the biofilm matrix. Olszak et al. (2019) observed that a specific Jumbo phage could significantly impact both planktonic and biofilm populations of *Pseudomonas aeruginosa*, leading to reduced biofilm formation and increased susceptibility to further treatments [146]. Furthermore, Gutiérrez et al. (2015) explored the anti-biofilm properties of the pre-neck appendage protein Dpo7 from phage vB_SepiS-phiIPLA7 in staphylococcal species. Their findings reveal that Dpo7 can effectively disrupt established biofilms and inhibit their formation. The ability of Dpo7 to target and degrade biofilm structures suggests a potential role in modulating the host’s innate immune response, particularly in biofilm-associated infections where the immune system struggles to clear biofilm-embedded bacteria. By facilitating the breakdown of biofilms, Dpo7 may enhance the accessibility of immune cells to bacterial cells, potentially influencing innate immune-signaling pathways and improving the overall efficacy of phage therapy against staphylococcal infections [150].

### 4.2. Advantages of Phage-Based Biofilm Treatments

Phage-based treatments have garnered significant attention as a promising solution for biofilm-related infections, offering several distinct advantages over traditional antimicrobial approaches. This section outlines the key benefits of phage therapy, emphasizing its specificity, self-replicating nature, and synergistic potential when combined with other antimicrobial agents. By leveraging these unique properties, phage therapy presents a targeted, sustained, and effective approach to addressing biofilm-associated infections, potentially transforming the landscape of bacterial infection management. Phage-based treatments offer several distinct advantages for addressing biofilm-related infections.

Specificity: Phages exhibit a high level of specificity, targeting particular bacterial species or strains without affecting beneficial microbiota. This specificity helps to minimize dysbiosis, a common side effect associated with broad-spectrum antibiotics. According to Pires et al. (2022), the precise targeting of phages reduces collateral damage and preserves the balance of the host’s microbiota, making phage therapy a more selective and potentially safer alternative to traditional antibiotics [153].

Self-replication: One of the key benefits of phage therapy is its auto-dosing capability. Phages replicate at the site of infection as long as there are susceptible bacteria present, providing sustained antibacterial activity. This self-replicating property decreases the need for frequent administration and ensures a prolonged therapeutic effect. Wang et al. (2024) highlight this advantage, noting that phages can continuously target and destroy bacteria within biofilms, which is particularly useful for chronic infections [154].

Synergistic effects: Phages can be combined with antibiotics or other antimicrobial agents to enhance overall treatment efficacy. This synergistic approach can help overcome bacterial resistance and reduce the likelihood of resistance development. Fedorov et al. (2023) discuss the successful application of phage–antibiotic combinations in treating periprosthetic infections, demonstrating how combining these treatments can improve outcomes and combat resistant bacterial strains [141]. Similarly, Ghanaim et al. (2023) emphasize the potential of phage therapy as a complementary strategy to antibiotics, especially for multidrug-resistant infections [142]. Abdelhamid and Yousef (2023) further review emerging strategies for combining phages with other antibiofilm agents, highlighting the potential for synergistic effects in managing persistent infections [143].

### 4.3. Case Studies and Applications

Phage therapy has demonstrated significant promise in managing various biofilm-related infections. Here are some illustrative case studies and applications.

Chronic wound infections: Chronic wounds, such as diabetic foot ulcers and pressure sores, often present biofilms that hinder healing. Phage therapy has shown promise in disrupting these biofilms and promoting wound healing [155]. Verbanic et al. (2022) highlighted the role of phages in chronic wound infections, emphasizing their effectiveness in targeting biofilm-forming bacteria and improving healing outcomes [156]. Additionally, Akturk et al. (2023) demonstrated that combining phages with antibiotics could enhance antibiofilm efficacy in an in vitro model, showing improved bacterial reduction and wound healing [144].

Cystic fibrosis (CF) lung infections: CF patients frequently suffer from chronic lung infections caused by biofilm-forming *Pseudomonas aeruginosa*. Conventional antibiotics often fail to eradicate these biofilms, leading to persistent infections and declining lung function. Fiscarelli et al. (2021) provided evidence of the effectiveness of phage therapy against *Pseudomonas aeruginosa* biofilms in cystic fibrosis patients, demonstrating significant biofilm disruption and bacterial killing [157]. Furthermore, Tan et al. (2021) reported successful personalized phage therapy for a carbapenem-resistant *Acinetobacter baumannii* lung infection in a patient with chronic obstructive pulmonary disease, illustrating the potential for tailored phage therapies in severe infections [158].

Medical device-associated infections: Biofilms on medical devices, such as catheters, prosthetic joints, and heart valves, pose significant risks due to their resistance to standard treatments. Phage therapy has been investigated as a method to prevent and treat these infections. For instance, Mirzaei et al. (2022) demonstrated that a phage cocktail could effectively control surface colonization by *Proteus mirabilis* in catheter-associated urinary tract infections, suggesting its potential for preventing device-related infections [159].

Dental plaque and periodontal disease: Dental biofilms, or plaque, are major contributors to dental caries and periodontal disease. Phage therapy has been explored as an alternative or adjunct to traditional plaque control methods. Kowalski et al. (2022) discussed the potential benefits of using bacteriophages in periodontal therapy, suggesting they could be an effective strategy for managing periodontal disease [160]. Chen et al. (2021) also highlighted the feasibility of phage therapy for periodontitis, showing that phages targeting dental biofilms could significantly reduce biofilm formation and improve oral hygiene [161].

### 4.4. Challenges and Future Directions

As phage therapy continues to emerge as a promising alternative for treating biofilm-associated infections, it is essential to critically address the obstacles that must be overcome to fully realize its potential. The following section explores the key challenges and future directions in the field, focusing on regulatory hurdles, phage resistance, and delivery mechanisms. By examining these issues, we aim to provide a comprehensive overview of the current landscape and highlight the necessary steps for advancing phage therapy from experimental applications to widespread clinical use.

Regulatory approval: Ensuring the safety, efficacy, and quality of phage preparations is crucial for gaining regulatory approval. Cooper et al. (2016) emphasize the need for standardized protocols and rigorous clinical trials to establish phage therapy as a viable therapeutic option [162]. Additionally, Pirnay et al. (2018) discuss the complexities of navigating regulatory pathways for phage-based therapeutics, underscoring the importance of developing clear guidelines [100].

Phage Resistance: Just as with antibiotics, bacteria can develop resistance to phages, which poses a significant challenge. Borin et al. (2021) highlight the importance of coevolutionary phage training to enhance bacterial suppression and delay the emergence of phage resistance [163]. This approach necessitates the continual discovery and development of new phages to stay ahead of bacterial adaptations. Recently, Kaneko et al. (2023) demonstrated that using a combination of bacteriophages with different physiological characteristics in a cocktail is crucial for effectively and continuously lysing bacteria, such as *Escherichia coli*, over a prolonged period while also suppressing the emergence of phage-resistant bacterial strains. This method shows promise as a strategy to enhance the efficacy of phage therapy [164].

Phage delivery: Effective delivery of phages to the biofilm site is critical for successful treatment. Kim et al. (2021) review advances in phage-delivering hydrogels, which can enhance phage stability and activity within biofilm environments, improving treatment outcomes [152]. Additionally, Malik et al. (2017) explore various formulation, stabilization, and encapsulation techniques for bacteriophages, which are essential for optimizing their delivery and efficacy [98].

In conclusion, while phage therapy offers a targeted approach to combating biofilm-associated infections, addressing these challenges will be key to realizing its full potential. By overcoming regulatory, resistance, and delivery issues, researchers and clinicians can further develop phage-based treatments to enhance patient outcomes and address the limitations of conventional antimicrobial strategies.

## 5. Phage Therapy for Intracellular Bacteria

Intracellular bacteria, such as *Mycobacterium tuberculosis*, *Salmonella* spp., and *Chlamydia trachomatis*, reside within host cells, making them challenging to target with conventional antibiotics. Phage therapy offers a promising approach to combatting these infections by exploiting the ability of bacteriophages to infect and replicate within host cells. This section explores the application of phage therapy for intracellular bacterial infections, including the mechanisms of action, advantages, and case studies demonstrating efficacy.

### 5.1. Mechanisms of Phage Action Against Intracellular Bacteria

In recent years, the therapeutic potential of bacteriophages, or phages, in combating bacterial infections has garnered significant attention. Phages have demonstrated remarkable efficacy in targeting and eliminating bacteria, even those residing within host cells. The following text delves into the mechanisms through which phages can target intracellular bacteria, highlighting their ability to penetrate host cells, induce bacterial lysis, and modulate host immune responses. By exploring these mechanisms, we gain insight into the promising role of phage therapy in treating intracellular bacterial infections. Phages can target intracellular bacteria through several mechanisms:

Intracellular penetration: Certain phages possess the ability to penetrate host cells and deliver their genetic material into the intracellular compartment, where they can replicate and produce progeny phages. This allows phages to target bacteria residing within host cells, such as macrophages or epithelial cells. For instance, Schmalstig et al. (2024) demonstrated that bacteriophages could infect and kill intracellular *Mycobacterium abscessus* within macrophages [165]. Similarly, Yang et al. (2024) reported the successful use of bacteriophage therapy in humanized mice infected with *Mycobacterium tuberculosis* [166].

Lysis of intracellular bacteria: Once inside the host cell, phages can induce the lysis of intracellular bacteria, leading to the release of phage progeny and the subsequent infection of neighboring bacterial cells. This process effectively reduces the intracellular bacterial load and promotes the clearance of infection. Johansen et al. (2021) illustrated that phage-antibiotic therapy could significantly enhance the clearance of drug-resistant *Mycobacterium abscessus* by promoting bacterial lysis [167].

Modulation of host immune response: Phages can modulate host immune responses to enhance bacterial clearance. For example, they can stimulate the production of pro-inflammatory cytokines or activate immune cells, such as macrophages, to facilitate bacterial killing. This immunomodulatory effect was noted by Dedrick et al. (2017), where bacteriophage therapy stimulated immune responses to combat *Mycobacterium tuberculosis* infections [168].

### 5.2. Advantages of Phage Therapy for Intracellular Bacterial Infections

The use of bacteriophages in treating bacterial infections has expanded beyond traditional applications to include targeting intracellular pathogens. This advancement is crucial in addressing infections that are otherwise difficult to treat with conventional antibiotics. The following text explores the specific mechanisms through which phages can effectively target intracellular bacteria, such as their ability to deliver genetic material directly into infected cells, their persistence within these cells to maintain antibacterial activity, and their synergistic interactions with host immune responses. These mechanisms underscore the potential of phage therapy in providing a targeted, effective approach to combating intracellular bacterial infections.

Targeted intracellular delivery: Phages can specifically target intracellular bacteria while sparing host cells, minimizing off-target effects, and reducing the risk of host cell damage. Shield et al. (2021) highlighted the specificity of bacteriophages in targeting mycobacterial infections without harming the host cells [169].

Intracellular persistence: Phages can replicate within host cells, ensuring sustained antibacterial activity at the site of infection and overcoming the limitations of conventional antibiotics, which may have limited intracellular penetration or activity. Dedrick et al. (2022) showed that phages could persist and replicate within infected cells, maintaining antibacterial efficacy [170].

Synergy with host immune responses: Phages can synergize with host immune responses to enhance bacterial clearance, potentially overcoming immune-evasion mechanisms employed by intracellular bacteria. This synergy was demonstrated early by Broxmeyer et al. (2002), where phage therapy enhanced the host’s immune response to clear intracellular *Mycobacterium tuberculosis* [171].

### 5.3. Case Studies and Applications

The application of phage therapy in treating intracellular bacterial infections offers a novel and promising approach to combating pathogens that reside within host cells. Traditional antibiotics often struggle with limited intracellular penetration and activity, making phage therapy an attractive alternative. The following case studies and applications highlight the potential of phage therapy in targeting intracellular bacteria such as *Mycobacterium tuberculosis*, *Salmonella* spp., and *Chlamydia trachomatis*, demonstrating significant reductions in bacterial load and enhanced treatment efficacy.

Tuberculosis (TB): *Mycobacterium tuberculosis*, the causative agent of TB, primarily infects macrophages, where it can persist and evade host immune responses. Phage therapy has shown promise in targeting intracellular *Mycobacterium tuberculosis*. A study by Schmalstig et al. (2024) demonstrated that mycobacteriophage could infect and replicate within *Mycobacterium tuberculosis*-infected macrophages, leading to a significant reduction in intracellular bacterial load [165].

*Salmonella* spp.: *Salmonella* infections are associated with a range of clinical manifestations, including gastroenteritis, typhoid fever, and systemic infections. The rise of antibiotic-resistant *Salmonella* strains has spurred interest in alternative treatment strategies, such as phage therapy. Recent studies have shown that bacteriophages can effectively target intracellular *Salmonella*, a challenging infection site for conventional antibiotics due to the bacteria’s ability to reside within host cells. In vitro experiments have demonstrated that specific phages can significantly reduce the intracellular *Salmonella* load, suggesting their potential as adjunctive therapies to enhance the efficacy of existing antibiotics and reduce the development of resistance [172,173]. Additionally, animal models have confirmed the ability of phages to penetrate and disrupt *Salmonella* within infected tissues, offering hope for treating persistent infections where traditional antibiotics may fall short [112]. This approach could be particularly valuable in cases of multidrug-resistant *Salmonella* strains where treatment options are increasingly limited [174]. Furthermore, combining phage therapy with antibiotics has shown synergistic effects, potentially lowering the required antibiotic dose and minimizing side effects [175].

*Chlamydia trachomatis: Chlamydia trachomatis* is a small, Gram-negative obligate intracellular pathogen that causes sexually transmitted infections and trachoma in humans. Due to the challenges posed by antibiotic resistance, phage therapy has emerged as a promising alternative for targeting such intracellular pathogens. Notably, a study has shown that with rising antibiotic-resistant *Chlamydia trachomatis* (CT) infections, the chlamydia-specific lytic phage ΦCPG1 has emerged as a promising treatment. ΦCPG1 has demonstrated broad inhibitory effects on all CT serotypes, effectively disrupting infection stages and inhibiting bacterial growth [176]. This phage’s ability to target CT highlights its potential as a novel therapeutic agent. Additionally, the engineered pGFP-ΦCPG1 phage offers a valuable tool for future research on CT drug resistance and vaccine development [176]. This finding aligns with ongoing research into chlamydiaphages, a group of bacteriophages known to infect various Chlamydia species, highlighting their potential as a novel therapeutic strategy in combating infections caused by these dangerous microbes [177].

### 5.4. Challenges and Future Directions

Despite the promising potential of phage therapy for treating intracellular bacterial infections, several challenges must be addressed to optimize its effectiveness. Key issues include ensuring an effective delivery of phages into host cells, balancing the therapy with the host immune response, and managing the emergence of phage resistance. The following discussion explores these challenges in detail, highlighting recent advancements and strategies to overcome them.

While phage therapy for intracellular bacterial infections holds great promise, several challenges must be addressed, First, intracellular delivery would be an issue. Developing effective delivery methods to ensure phage penetration and replication within host cells is crucial for the success of intracellular phage therapy. Beitzinger et al. (2021) suggested using dendritic mesoporous silica nanoparticles to enhance phage delivery and activity against intracellular *Mycobacterium tuberculosis* [178]. Secondly, the host immune response also needs to be discussed. Understanding the interplay between phages, intracellular bacteria, and host immune responses is essential for optimizing therapeutic outcomes and minimizing adverse effects. Nick et al. (2022) emphasized the importance of balancing phage therapy with immune modulation to prevent adverse immune reactions [179]. Lastly, phage resistance is another of the most considered issues. Monitoring and mitigating the emergence of phage resistance in intracellular bacteria is critical for the long-term efficacy of phage therapy. Guerrero–Bustamante et al. (2021) discussed strategies to develop phage cocktails to overcome resistance and enhance therapeutic success [180].

In conclusion, phage therapy represents a promising approach for targeting intracellular bacterial infections. By harnessing the unique properties of phages, researchers and clinicians can develop innovative treatments to overcome the challenges posed by intracellular bacteria and improve patient outcomes.

## 6. Phage-Based Vaccines

Phage-based vaccines represent a cutting-edge approach to vaccine development, offering distinct advantages over traditional platforms. This section explores the principles underlying phage-based vaccine design, including the use of phages as carriers for antigen delivery, the efficacy of phage display systems, and the benefits of multivalent and adjuvant properties. By integrating antigenic peptides into phage structures, these vaccines can enhance immune responses and provide broad protection against various pathogens. The innovative design and versatile applications of phage-based vaccines underscore their potential to revolutionize vaccine development and address a wide range of infectious diseases. Figure 3 illustrates the overall mechanism of immune response induction by phage-based vaccines.

### 6.1. Principles of Phage-Based Vaccine Design

Phage-based vaccines utilize bacteriophages as carriers to deliver immunogenic epitopes, capitalizing on their ability to present antigenic peptides or proteins to the immune system effectively. This section outlines the core principles of phage-based vaccine design, including the role of phage-display systems in presenting antigens, the advantages of multivalent vaccines in targeting diverse pathogens, and the intrinsic adjuvant properties of phages that enhance immune responses. By integrating these principles, phage-based vaccines offer innovative and versatile solutions for developing effective vaccines against various infectious diseases.

Display Systems: Phage-display systems, including filamentous phages such as M13 and T7 phages, are instrumental in presenting foreign antigens on the phage surface. Antigenic peptides or proteins are genetically fused to phage coat proteins, which are then expressed and displayed during phage replication. This method has been extensively reviewed by González–Mora et al. (2020) and Mohammad Hasani et al. (2023), highlighting its efficacy in antigen delivery [11,185].

Multivalent Vaccines: Phage-based vaccines can present multiple antigenic epitopes simultaneously, thereby enhancing the breadth and specificity of the immune response. Such multivalent vaccines can target diverse strains or variants of a pathogen, offering broader protection. For instance, Bao et al. (2019) emphasize the versatility of multivalent phage-based vaccines in providing extensive coverage against antigenically diverse pathogens [186]. Similarly, the application of this approach in vaccines for SARS-CoV-2 has been explored by Zhu et al. (2022) and Tao et al. (2018) [187,188].

Adjuvant Properties: Phages exhibit intrinsic adjuvant properties, capable of stimulating innate immune responses. They can activate Toll-like receptors (TLRs) and other pattern-recognition receptors (PRRs), thereby enhancing antigen presentation and cytokine production. This adjuvant activity is crucial for the effectiveness of phage-based vaccines. Studies by Jepson and March (2004) and Górski et al. (2012) underscore the potential of phages to act as adjuvants in vaccine formulations [189,190]. The recent work by Krut and Bekeredjian–Ding (2018) further explores how phage therapy can modulate immune responses [191].

Phage-based vaccines are not only innovative in their design but also versatile in their applications. Their ability to present multiple antigens, coupled with their adjuvant properties, makes them a valuable tool in the development of vaccines against a wide range of infectious diseases.

### 6.2. Mechanisms of Immune Stimulation

Phage-based vaccines are emerging as promising tools in immunotherapy due to their ability to stimulate both innate and adaptive immune responses through diverse mechanisms. This section explores these mechanisms, highlighting the interactions between phages and the immune system that led to effective immunity. Specifically, it examines how phages activate innate immune responses, facilitate antigen presentation, and stimulate both T cell and B cell activation. The intricate processes involved and the potential of phage-based vaccines in advancing vaccine development are underscored by key studies in the field.

Innate Immune Activation: Phages initiate innate immune responses by interacting with pattern-recognition receptors (PRRs) on antigen-presenting cells (APCs), recognizing pathogen-associated molecular patterns (PAMPs). This interaction triggers the production of pro-inflammatory cytokines, chemokines, and Type I interferons, which enhance antigen presentation and immune cell recruitment. The intricate mechanisms of this activation are detailed by Van Belleghem et al. (2018) and Popescu et al. (2021), who highlight the role of phages in modulating the innate immune system [192,193]. Carroll–Portillo and Lin (2019) provide crucial insights into how bacteriophages can influence the innate immune-signaling pathway. Their study reveals that phages are capable of interacting with immune cells, such as macrophages and dendritic cells, and they can modulate key signaling pathways, including the Toll-like receptor (TLR) pathways. This interaction can lead to either the activation or suppression of innate immune responses, depending on the context. The ability of phages to modulate these pathways suggests that they may play a role in shaping the host’s immune environment, with potential implications for both therapeutic applications and the understanding of host–pathogen dynamics [181].

Antigen Presentation: Phage-based vaccines effectively deliver antigenic epitopes to APCs such as dendritic cells, macrophages, and B cells. These cells process and present the antigens to T cells, initiating the adaptive immune response. This process leads to the activation and differentiation of antigen-specific T cells and B cells. The efficacy of this antigen delivery and presentation is demonstrated by the work of Sartorius et al. (2015), who showed that filamentous bacteriophages can trigger robust immune responses through TLR9-mediated pathways [194]. Xu et al. (2022) also emphasize the utility of phage-display systems in targeting specific immune cells [195].

T Cell Activation: Phage-presented antigens are crucial for the activation of antigen-specific T cells. This activation results in the proliferation and differentiation of effector T cells. CD4+ T cells assist B cells in antibody production, while CD8+ T cells are instrumental in mediating cellular immunity by eliminating infected cells. The role of phage-based vaccines in T cell activation is explored by Chatterjee and Duerkop (2018), who discuss emerging paradigms in phage–eukaryotic host interactions [196].

B Cell Activation: Phage-based vaccines stimulate B cell activation and antibody production against the presented antigens. Activated B cells undergo clonal expansion and differentiate into plasma cells, which produce antigen-specific antibodies. These antibodies neutralize pathogens and facilitate their clearance from the body. Research by Eriksson et al. (2009) illustrates how phages can induce immune responses that lead to effective pathogen neutralization [197]. Additionally, Ragothaman and Yoo (2023) review advances in engineered phage-based vaccines, including their impact on B cell activation [198].

The multifaceted mechanisms by which phage-based vaccines stimulate both innate and adaptive immune responses underscore their potential as powerful tools in vaccine development and immunotherapy.

### 6.3. Applications of Phage-Based Vaccines

Phage-based vaccines present a versatile and innovative approach to combating infectious diseases and beyond. This section delves into the wide-ranging applications of these vaccines, highlighting their efficacy in targeting bacterial and viral pathogens, their potential in cancer immunotherapy, and their promise in addressing emerging infectious diseases. By leveraging the unique properties of phages, researchers are developing next-generation vaccines that stimulate robust immune responses and offer protection against diverse health threats. Key studies demonstrate the effectiveness and adaptability of phage-based vaccines, showcasing their potential to revolutionize vaccine development and public health interventions.

Phage-based vaccines offer a wide range of applications in infectious disease prevention and beyond. Their versatility makes them valuable tools in several areas:

Bacterial Vaccines: Phage-based vaccines have been successfully developed to combat various bacterial pathogens. For example, vaccines targeting *Streptococcus pneumoniae*, *Staphylococcus aureus*, and *Escherichia coli* utilize phages to display bacterial surface antigens, toxins, or virulence factors. This approach stimulates protective immune responses that help prevent colonization and infection. Tao et al. (2013) demonstrate the use of phage nanoparticles for developing next-generation plague vaccines, showcasing their efficacy in targeting bacterial pathogens [54]. Yang et al. (2006) highlight the potential of phage-based vaccines in substituting complex vaccine systems for bacterial infections [199].

Viral Vaccines: Phage-based vaccines are also promising for viral diseases such as influenza, HIV, and hepatitis. Engineered phages can present viral antigens to induce both humoral and cellular immune responses. This strategy has the potential to enhance vaccine efficacy and provide cross-protection against diverse viral strains. For instance, Shi et al. (2018) describe how phage vaccines displaying specific epitopes can protect against systemic candidiasis, illustrating their potential in viral and fungal infections [57]. Similarly, Gong et al. (2023) discuss the use of phage-display technology combined with epitope design to generate robust antibody responses against emerging pathogens like Tilapia Lake Virus [200].

Cancer Vaccines: Phage-based vaccines are emerging as innovative tools in cancer immunotherapy. By displaying tumor-associated antigens on phage surfaces, these vaccines can stimulate anti-tumor immune responses. The aim is to activate cytotoxic T cells and promote tumor regression. Studies such as those by Iwagami et al. (2017) demonstrate the potential of lambda phage-based vaccines in inducing antitumor immunity in hepatocellular carcinoma [201]. Tao et al. (2018) also explore phage T4 nanoparticles for creating dual vaccines against anthrax and plague, showcasing their versatility [188].

Emerging Infectious Diseases: Phage-based vaccines hold significant promise for rapid responses to emerging infectious diseases, including COVID-19. Phages can be engineered to display epitopes from novel pathogens, enabling quick design and production of vaccines to address new public health threats. For example, Staquicini et al. (2021) present targeted phage-based COVID-19 vaccination strategies with a streamlined cold-free supply chain, demonstrating their potential for addressing global health emergencies [183]. Ul Haq et al. (2023) emphasize the use of phage-based platforms for designing multiplex vaccines against COVID-19 [202].

These diverse applications highlight the adaptability and potential of phage-based vaccines in various fields, from combating infectious diseases to advancing cancer immunotherapy and addressing emerging health threats.

### 6.4. Challenges and Future Directions

Phage-based vaccines hold significant promise, yet their broader acceptance and application hinge on overcoming several challenges. This section addresses these challenges, focusing on the need to optimize antigen presentation to enhance immunogenicity, ensure safety by assessing potential adverse effects, and navigate the regulatory landscape to achieve approval. Additionally, it explores future directions, emphasizing advancements in technology, evolving regulatory frameworks, and the necessity for clinical trials to validate the efficacy and safety of phage-based vaccines. By addressing these challenges and leveraging future advancements, phage-based vaccines can realize their full potential in diverse applications.

Phage-based vaccines show considerable potential but face several challenges that need addressing for broader acceptance and application:

Immunogenicity: Optimization of phage display systems and antigen presentation strategies is needed to enhance vaccine immunogenicity and efficacy. Enhancing the effectiveness of phage-based vaccines involves improving how antigens are presented. As Zalewska–Piątek (2023) note, the optimization of phage-display systems is crucial for boosting the immunogenicity of these vaccines and improving their efficacy in diverse applications. The effectiveness of these vaccines hinges on refining antigen presentation strategies to ensure a robust immune response [117].

Safety: Evaluation of potential adverse effects, such as phage-mediated immune activation or autoimmunity, is essential for vaccine safety assessment. Ensuring safety is critical for the success of phage-based vaccines. Henein (2013) highlights concerns about potential adverse effects, such as phage-mediated immune activation or autoimmunity, which necessitate thorough evaluation [203]. Addressing these safety concerns through rigorous testing and validation is essential to prevent unforeseen negative outcomes.

Regulatory Approval: Standardization of manufacturing processes and validation of vaccine efficacy are required for regulatory approval and widespread deployment. Achieving regulatory approval involves standardizing manufacturing processes and validating vaccine efficacy. Furfaro et al. (2018) emphasize the regulatory hurdles faced by phage therapy and the need for standardized protocols to ensure consistency and safety in clinical applications [204]. Similarly, Verbeken et al. (2012) discuss the challenges in optimizing regulatory frameworks, particularly in Europe, for sustainable phage therapy [205].

Advancements and Regulation: Moving forward, the development of phage-based vaccines will benefit from advancements in technology and regulatory practices. The European regulatory framework, as discussed by Faltus (2024), is evolving to better accommodate medicinal phages, which may streamline the approval process and enhance the adoption of these vaccines [206]. Additionally, Strathdee et al. (2023) emphasize the need for continued research into the biological mechanisms of phages and their potential future applications [207].

Clinical Trials and Practical Applications: As noted by Manohar et al. (2019), there is a need for more clinical trials to better understand the pharmacological and immunological aspects of phage therapy, which will be pivotal in overcoming current limitations and demonstrating practical benefits [208].

In conclusion, while phage-based vaccines offer a promising approach to infectious disease prevention, overcoming these challenges is essential for their successful implementation. By addressing immunogenicity, safety, and regulatory issues, researchers and developers can enhance the efficacy and acceptance of these innovative vaccines.

## 7. Phage Therapy as Anti-Cancer Agents

Phage therapy has gained attention as a promising approach in cancer treatment, leveraging the unique properties of bacteriophages to specifically target and destroy tumor cells. This section explores the mechanisms behind phage-mediated anti-cancer effects, the progress in developing phage-based cancer therapies, and their potential applications. Table 5 provides an overview of the phages utilized in cancer theranostics, highlighting their therapeutic potential in this emerging field.

### 7.1. Mechanisms of Phage-Mediated Anti-Cancer Activity

Phages have emerged as promising anti-cancer agents due to their ability to target and kill cancer cells through various mechanisms. This section explores the diverse ways in which phages exert their anti-cancer effects, including their capacity for tumor targeting, direct cytotoxicity, and indirect modulation of the tumor microenvironment. By engineering phages to specifically bind to tumor cells, induce cell death, and stimulate immune responses, researchers are advancing novel cancer therapies that leverage the unique properties of phages. Key studies highlight the potential of phage technology in enhancing cancer detection and treatment, showcasing its versatility and efficacy in oncology.

Tumor Targeting: Phages can be engineered to specifically target tumor cells by displaying tumor-specific ligands or peptides on their surfaces. This targeting capability is facilitated through ligands that bind to receptors overexpressed on cancer cells, promoting selective phage internalization and subsequent tumor cell destruction [225]. For instance, Veeranarayanan et al. (2021) highlight the advancement of phage technology in targeting solid tumors by conjugating phages with molecules that recognize tumor-associated antigens, enhancing the specificity and efficacy of treatment [13].

Direct Tumor Cell Killing: Upon attachment to tumor cells, phages can induce direct cytotoxic effects through several mechanisms. These include the release of phage-encoded cytotoxic proteins or enzymes, activation of apoptotic pathways, and disruption of essential cellular processes [226]. Przystal et al. (2019) demonstrated the effectiveness of systemic temozolomide-activated phage-targeted gene therapy in human glioblastoma, illustrating how phages can be engineered to deliver therapeutic genes directly to tumor cells, leading to their destruction [209].

Indirect Anti-Tumor Effects: Beyond direct cytotoxicity, phages can exert indirect anti-tumor effects by modulating the tumor microenvironment and stimulating immune responses against cancer cells. They can activate immune cells, such as dendritic cells and T cells, thereby enhancing the recognition and elimination of tumor cells by the immune system [227]. For example, Fredrik Eriksson et al. (2009) demonstrated that tumor-specific bacteriophages could induce tumor destruction through the activation of tumor-associated macrophages, highlighting the potential of phages to orchestrate an immune-mediated attack on tumors [197].

Additionally, the development of modified bacteriophages for tumor detection and targeted therapy has been explored. Shen et al. (2023) reported on the use of engineered phages for enhanced tumor detection and targeted treatment, showcasing advances in phage technology that improve the precision and efficacy of cancer therapy [14]. Similarly, Dong et al. (2020) discussed the use of bioinorganic hybrid bacteriophages to modulate the intestinal microbiota and remodel the tumor-immune microenvironment against colorectal cancer, emphasizing the versatility of phage-based approaches in cancer treatment [28].

### 7.2. Development of Phage-Based Cancer Therapies

Phage-based cancer therapies are advancing the frontiers of cancer treatment by harnessing the unique capabilities of phages. This section reviews innovative strategies in this field, including targeted drug delivery, gene therapy, and immunotherapy. Phages offer a versatile platform for precisely delivering therapeutic agents to tumor cells, encoding therapeutic genes for direct cancer cell targeting, and stimulating anti-tumor immune responses. Key developments in these areas highlight the potential of phage-based approaches to improve the specificity, efficacy, and versatility of cancer treatments. The advancements in targeted delivery systems, gene therapies, and immune modulation underscore the promising future of phage technology in oncology.

Targeted Drug Delivery: Phages offer a promising approach for targeted drug delivery to tumor cells, enhancing the efficacy and specificity of anti-cancer treatments. Drugs or therapeutic agents can be conjugated to phage surfaces or encapsulated within phage capsids, allowing for precise delivery to tumor sites [228]. For example, Kumar et al. (2023) describe the use of phages as vehicles for delivering therapeutic agents, improving the targeted delivery of drugs to cancerous tissues [229]. Similarly, Wang et al. (2024) review various phage-based delivery systems, highlighting their applications and challenges in nanomedicines for targeted cancer therapy [16].

Gene Therapy: Phages can be engineered to deliver therapeutic genes directly to tumor cells, presenting a novel approach for cancer gene therapy. Phage genomes can be modified to incorporate gene expression cassettes that encode anti-cancer proteins, immunomodulatory factors, or suicide genes, facilitating targeted gene delivery to tumor cells [210]. Sittiju et al. (2024) have developed bacteriophage-based particles carrying the TNF-related apoptosis-inducing ligand (TRAIL) gene for targeted delivery in hepatocellular carcinoma, demonstrating the potential of phages in gene therapy [211]. Furthermore, Al-Bahrani et al. (2023) utilized a transmorphic phage-guided systemic delivery of the TNFα gene for treating pediatric medulloblastoma, showcasing the versatility of phage-mediated gene therapy [212].

Immunotherapy: Phages can also function as immunotherapeutic agents to stimulate anti-tumor immune responses. Phage-display libraries can be used to identify tumor-specific antigens or epitopes, which can then be used to generate phage-based vaccines or immunomodulatory agents for cancer immunotherapy [213]. For example, Hajitou (2010) discusses the role of targeted systemic gene therapy and molecular imaging in cancer, highlighting the potential of phage vectors for enhancing immune responses against tumors [230]. Moreover, Dong et al. (2023) explored a self-adjuvanting phage-enabled hydrogel for remodeling the tumor microenvironment, demonstrating the potential of phages in immune modulation and cancer therapy [214].

### 7.3. Applications of Phage-Based Cancer Therapies

Phage-based cancer therapies are emerging as a transformative approach with a range of applications in cancer treatment. This section explores their potential across different cancer types, including solid tumors, hematological malignancies, and combination therapies. Engineered phages have demonstrated efficacy in targeting and treating solid tumors, enhancing the effectiveness of treatments for cancers such as breast, lung, and colon. In hematological malignancies, phages offer novel strategies for gene delivery and immunomodulation. Additionally, integrating phage-based therapies with conventional treatments like chemotherapy and radiation could improve overall outcomes by enhancing tumor cell destruction and reducing resistance. Advances in these areas underscore the significant promise of phage technology in advancing cancer treatment strategies

Solid Tumors: Phage-based therapies have been explored for treating solid tumors such as breast cancer, lung cancer, and melanoma. Engineered phages displaying tumor-targeting ligands or cytotoxic payloads have shown efficacy in preclinical models of these solid tumors [231]. For instance, Turrini et al. (2024) engineered a spheroid-penetrating phage nanovector for the photodynamic treatment of colon cancer cells, illustrating the application of phage-based systems in solid tumor therapy [232].

Hematological Malignancies: Research is also ongoing into phage-based therapies for hematological malignancies like leukemia and lymphoma. Phage-mediated gene delivery and immunomodulation strategies have the potential to enhance the efficacy of existing therapies and address issues of drug resistance [212]. Zheng et al. (2019) demonstrated that phage-guided modulation of the gut microbiota can augment responses to chemotherapy in colorectal cancer models, highlighting a novel application of phage therapy in hematological malignancies [215]. Additionally, phage therapy shows promise in addressing complications like acute graft-versus-host disease (aGVHD) after allogeneic hematopoietic cell transplantation (allo-HCT). Gut dysbiosis, particularly the overgrowth of *Enterococcus faecalis,* is a known risk factor for aGVHD. Recent research by Fujimoto et al. (2024) has identified a phage-derived enzyme targeting *E. faecalis* biofilms, which are resistant to conventional antibiotics. In aGVHD-induced mice, this enzyme reduced *E. faecalis* levels and improved survival, highlighting its potential as a novel therapeutic strategy for protecting against aGVHD and improving outcomes in hematological malignancies [233].

Combination Therapies: Integrating phage-based therapies with conventional treatments such as chemotherapy, radiation therapy, and immune checkpoint inhibitors holds significant potential to enhance overall treatment outcomes. Such combinations may lead to improved tumor cell eradication, reduced treatment resistance, and minimized adverse effects [234]. For instance, the photothermal phage developed by Shahri–Varkevishahi et al. (2021) demonstrates the utility of virus-based agents in photothermal therapy, which can be further combined with other treatment modalities for heightened efficacy [235]. Moreover, phage application in modulating gut microbiota, particularly by targeting colorectal cancer-associated bacteria such as enterotoxigenic *Bacteroides fragilis* (ETBF), offers promising therapeutic potential [236]. These emerging strategies underscore the multifaceted role of phages in expanding the therapeutic arsenal against cancer.

### 7.4. Challenges and Future Directions

Despite their promising potential, phage-based cancer therapies face several challenges that must be addressed to fully realize their benefits. This section examines key issues, including enhancing the specificity and efficiency of tumor targeting, managing immune responses to avoid adverse effects, and navigating the transition from preclinical research to clinical trials. Improving targeting mechanisms, understanding and modulating immune interactions, and ensuring rigorous safety and efficacy evaluations are essential for advancing phage-based therapies. Addressing these challenges through ongoing research and development will be crucial for optimizing the application of phage technology in cancer treatment.

Tumor Targeting: Enhancing the specificity and efficiency of phage targeting to tumor cells while minimizing off-target effects is crucial. Improved targeting mechanisms are needed to ensure that phages can accurately home in on cancer cells without affecting healthy tissues. Advances in phage engineering, such as the development of tumor-specific ligands or improved phage display techniques, are essential for overcoming this challenge [237]. Wang et al. (2024) discuss the engineering and applications of phage-based delivery systems, including strategies to improve targeting precision and reduce off-target effects [16].

Immune Response: The interaction between phage-based therapies and the host immune system needs to be thoroughly understood and managed. Phage therapies can elicit immune responses that may affect treatment efficacy and lead to adverse effects. Strategies to modulate the immune response, such as using immuno-evasive phages or combining phage therapy with immune modulation, are critical for optimizing treatment outcomes [238]. Li et al. (2023) highlight recent progress in phage-based nanoplatforms for tumor therapy, including considerations for immune system interactions and strategies for minimizing immune-related issues [239].

Clinical Translation: Transitioning phage-based cancer therapies from preclinical studies to clinical trials involves rigorous evaluation of safety, efficacy, and pharmacokinetics in human subjects. This process requires comprehensive clinical trials to ensure that phage therapies are safe and effective for human use. Ensuring consistent manufacturing, understanding potential side effects, and establishing appropriate dosing regimens are all critical steps in advancing these therapies [240]. Petrenko and Gillespie (2017) review the paradigm shift in bacteriophage-mediated delivery of anticancer drugs and the steps needed for successful clinical translation [237].

In conclusion, phage therapy presents a promising approach to cancer treatment, offering targeted and multifaceted strategies to address various types of cancer. Continued research and development are essential for overcoming current challenges and improving the efficacy and safety of phage-based cancer therapies. With advancements in phage engineering, immune modulation, and clinical evaluation, phage-based therapies have the potential to make a significant impact on cancer treatment outcomes and enhance patient survival.

## 8. Phages as Drug Delivery Systems (DDS)

Phages, with their unique biological characteristics, have emerged as powerful tools in drug delivery, offering innovative solutions to enhance therapeutic precision and efficacy. This section provides a detailed examination of phage-based drug delivery systems, exploring their underlying principles, advancements in vector engineering, and a wide array of medical applications. By addressing the fundamental aspects of phage functionality, engineering innovations, and diverse therapeutic uses, as well as examining the challenges and future directions, this overview underscores the potential of phages to revolutionize targeted drug delivery and improve patient outcomes across various medical fields.

Considering the application of phages as therapeutic agents or delivery vehicles in clinical settings, it is crucial to recognize that free phages may be susceptible to various stresses and environmental factors. To maximize their efficacy, protective measures and concentration strategies for phages are essential. As illustrated in Figure 4, several representative methods for phage modification and drug-delivery system (DDS) applications are presented.

Encapsulating phages within liposomes, niosomes, or hydrogels in an aqueous phase offers protection against environmental stresses such as physicochemical stressors and immunological responses (Figure 4(Ia)) [241,242]. Additionally, to enhance the concentration of phages within the reaction system, they can be immobilized using nanocarriers or mesoporous nanoparticles, which helps localize and maintain an effective phage concentration at the target site (Figure 4(Ib)) [243].

While phages can inherently function as antimicrobial agents by lysing their host bacteria, their therapeutic potential can be further enhanced by adding other functional elements. As shown in Figure 4II, phages can be engineered to carry additional functions. For instance, functional proteins such as enzymes or peptide antigens can be expressed on the phage capsid or at the tips of long tail fibers. In filamentous phages, these proteins can be expressed on major coat proteins. Moreover, through genetic engineering techniques, mRNA or DNA can be encapsulated within the phage capsid, expanding the utility of phages beyond simple antibacterial action to include roles in vaccine development and gene therapy [241,243,244].

**Figure 4 antibiotics-13-00870-f004:**
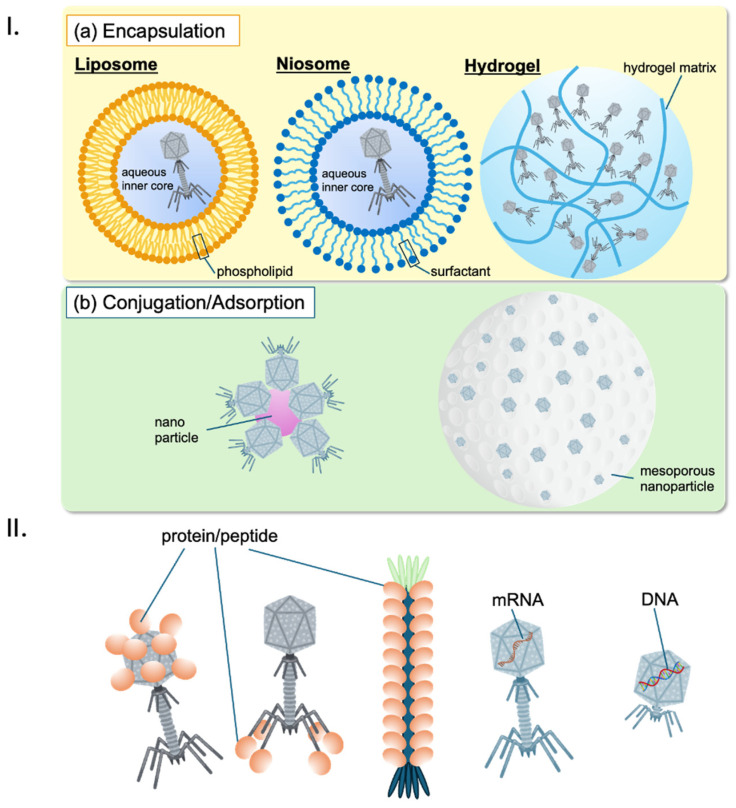
Phage-based drug-delivery systems: (**I**). Immobilization of phages: (**a**) Encapsulation of phage in liposomes (**left**) [182,245], niosomes (**middle**) [77,246] and in hydrogel beads (**right**) [247]; (**b**) conjugation/adsorption of phages onto nanoparticles (**left**) [248] and mesoporous nanoparticles (**right**) [249], respectively. These encapsulation methods protect phages from physicochemical stresses (e.g., pH, shear stress) and/or immunological reactions (e.g., phagocytosis, complement-mediated neutralization). (**II**) Possible drugs delivered by phages as vectors: Proteins (e.g., enzymes, antigens) or peptides can be expressed on the phage capsid, the tips of long tail fibers in Caudovirales, or the major coat proteins in filamentous phages. Additionally, mRNA and/or DNA-encoding targeted genes can be loaded into the phage capsid.

### 8.1. Principles of Phage-Based Drug Delivery

Phage-based drug-delivery systems represent a cutting-edge approach that leverages the unique properties of bacteriophages to enhance therapeutic efficacy. By exploiting their natural affinity for specific target cells and their ability to be engineered with precision, phages offer a versatile platform for drug delivery. This section explores the fundamental principles underlying phage-based drug-delivery systems, highlighting key aspects such as targeting specificity, payload capacity, and stability. These attributes collectively position phages as promising vehicles for advancing therapeutic interventions.

Targeting Specificity: Phages can be engineered to present specific targeting ligands or peptides on their surface, allowing for precise delivery to targeted cell types or tissues. This strategy utilizes the ability of phages to bind to receptors overexpressed on target cells, facilitating both internalization and payload delivery. For instance, Zhao et al. (2024) discuss how bacteriophage proteins can guide antibiotics towards their intended targets, enhancing the specificity and efficacy of the treatment [250].

Payload Capacity: Phage capsids can accommodate a wide range of therapeutic agents, including small molecules, peptides, proteins, nucleic acids, and nanoparticles. This versatility is crucial for delivering multiple therapeutic agents simultaneously. Aljabali et al. (2023) highlight the diverse payloads that phages can carry, underscoring their potential in developing multi-functional therapeutic systems [251].

Stability and Protection: Phage capsids offer a protective environment for encapsulated payloads, shielding them from degradation and enzymatic activity in the extracellular space. This property ensures that therapeutic agents reach their target sites effectively while minimizing off-target effects. Singla et al. (2016) demonstrated that encapsulating bacteriophages in liposomes enhances their stability and entry into macrophages, further protecting them from neutralizing antibodies [252].

### 8.2. Engineering of Phage Vectors for Drug Delivery

The evolution of phage vector engineering has significantly advanced the field of drug delivery, offering innovative strategies to improve therapeutic outcomes. By modifying phage vectors to incorporate surface-display systems, encapsulation techniques, and modular design, researchers are pushing the boundaries of what these biological entities can achieve. This section delves into the latest engineering approaches that enhance the functionality and effectiveness of phage-based drug delivery systems, underscoring their potential in addressing complex therapeutic challenges.

Surface Display Systems: Phages can be modified to display targeting ligands, therapeutic peptides, or antibodies on their surface. Phage-display libraries, as discussed by Kumar et al. (2023), enable the selection of high-affinity ligands for specific cell types or disease markers, facilitating targeted drug delivery [229].

Encapsulation Strategies: Phages can encapsulate therapeutic payloads using various methods, including chemical conjugation, genetic fusion, or self-assembly techniques. Naskalska and Heddle (2024) review how virus-like particles derived from bacteriophage MS2 serve as antigen scaffolds and protective shells, emphasizing their role in encapsulating and delivering therapeutic agents [253].

Modularity and Flexibility: Phage vectors offer modularity in payload design and delivery, with genetic engineering techniques allowing for customizable cargo sequences. This flexibility enables the creation of tailored drug-delivery systems for specific therapeutic needs. Ju and Sun (2017) discuss how filamentous bacteriophages and phage-mimetic nanoparticles can be employed as versatile drug-delivery vectors [254].

### 8.3. Applications of Phage-Based Drug Delivery Systems

Phage-based drug-delivery systems are revolutionizing various areas of medicine by offering targeted and effective treatment solutions. Their unique properties make them suitable for a range of applications, from cancer therapy and gene therapy to vaccine delivery and treatment of infectious diseases. This section reviews the diverse applications of phage-based systems, emphasizing their potential to improve therapeutic precision and address challenges associated with conventional treatment methods. Phage-based drug-delivery systems offer a wide range of medical applications, including the following aspects.

Cancer Therapy: Phage vectors are being explored for delivering chemotherapeutic agents, targeted therapies, and nucleic acid-based therapeutics to cancer cells with increased specificity and efficacy. Garg (2019) highlights the potential of filamentous bacteriophages in targeted cancer therapy, which minimizes systemic toxicity and enhances therapeutic outcomes [255].

Vaccine Delivery: Phage-based systems can enhance vaccine efficacy by displaying antigens or adjuvants on their surface. This strategy promotes antigen uptake and immune activation. As reported by Sittiju et al. (2024), phage-based particles carrying therapeutic genes are being investigated for targeted delivery in cancer therapy, which could be adapted for vaccine applications [211].

Infectious Disease Treatment: Phage vectors offer potential solutions for delivering antimicrobial agents or antiviral drugs to specific pathogens. This approach could be particularly useful in combating drug-resistant infections. Nieth et al. (2015) discuss the use of liposome-mediated intracellular bacteriophage therapy as a novel strategy for treating bacterial infections [256].

### 8.4. Challenges and Future Directions

While phage-based drug-delivery systems offer significant advantages, several challenges must be addressed to fully realize their potential. Key issues such as immunogenicity, tumor penetration, and clinical translation need careful consideration. Despite their benefits, these systems face several hurdles. This section explores these challenges and outlines future directions for overcoming them, emphasizing the importance of ongoing research and development in advancing phage-based therapies.

Immunogenicity: Phage vectors may provoke immune responses in vivo, which can limit their efficacy and safety. Addressing immunogenicity through modifications and enhancing biocompatibility is crucial for clinical application. Hathaway et al. (2017) emphasize the need for continued research to mitigate immunogenicity issues [257].

Tumor Penetration: Enhancing the penetration and distribution of phage-based drug-delivery systems within solid tumors remains a significant challenge. Improving tumor targeting and overcoming tumor heterogeneity are critical for maximizing therapeutic efficacy. et al. (2013) discuss various nano/micro formulations that could improve phage delivery in cancer therapy [241].

Clinical Translation: Moving phage-based drug-delivery systems from preclinical studies to clinical trials requires a thorough evaluation of safety, pharmacokinetics, and efficacy. This step is essential for ensuring the viability of phage-based therapies in human subjects. Rastogi et al. (2017) provide insights into the development and clinical translation of phage-based therapies, highlighting the importance of rigorous testing [258].

In summary, phages present a versatile and effective platform for drug delivery, with broad applications across medicine and biotechnology. Continued research and development are likely to advance phage-based drug delivery systems, potentially transforming targeted therapeutics and improving patient outcomes.

## 9. Phage-Display Technology in Drug Discovery

Phage-display technology leverages bacteriophages to present peptide or protein libraries on their surfaces, enabling the rapid identification of high-affinity ligands for therapeutic purposes. This section highlights its role in drug discovery, including target identification, lead optimization, and antibody and peptide drug development.

### 9.1. Applications in Drug Discovery

Phage display has revolutionized drug discovery, particularly in screening for target molecules, optimizing therapeutic leads, and engineering antibodies. By showcasing advancements and case studies, this section emphasizes the transformative impact of this technology on therapeutic development.

Target Identification: Phage libraries allow for the screening of ligands against target proteins. Studies like those by França et al. (2023) illustrate how phage display identifies novel cancer immunotherapy targets, facilitating the discovery of new therapeutic leads [259].

Lead Optimization: Phage display enables the optimization of selected ligands through rounds of mutagenesis and selection. For example, Miki et al. (2022) demonstrate the improvement of stapled peptide ligands, highlighting the technology’s potential to enhance drug candidates’ efficacy [260].

Antibody Engineering: Phage display plays a pivotal role in designing antibodies with enhanced properties such as higher affinity and specificity. Wang et al. (2019) describe the engineering of bispecific fusion proteins against MERS-CoV, a testament to the technology’s utility in antibody engineering [261].

Peptide Drug Discovery: This technology also supports the identification of peptides with therapeutic potential. Guerlavais et al. (2023) showcase the development of stabilized α-helical peptides for drug discovery, reinforcing the value of phage display in the peptide domain [262].

### 9.2. Recent Advances in Phage Display for Drug Discovery

Advances in next-generation sequencing (NGS), multi-modal screening, and bioinformatics have significantly enhanced the precision and efficiency of phage display. The integration of these technologies allows for better analysis of phage libraries, optimized ligand screening, and deeper insights into antibody diversity. For example, NGS enables large-scale sequencing of antibody repertoires, providing key information on ligand diversity and evolution [263].

NGS Integration: This has allowed for a more in-depth analysis of selected phages, leading to the discovery of rare, high-affinity ligands [264].

Multi-Modal Screening: Phage display now integrates cell-based assays and protein–protein interaction assays, expanding the diversity and quality of identified ligands. For example, André et al. (2022) discuss using in vivo phage display for improving antibody targeting and drug-delivery properties [265].

Bioinformatics: Advanced algorithms have improved phage-display data analysis, facilitating the design of more effective therapeutic antibodies [266].

### 9.3. Challenges and Future Directions

Despite its success, challenges remain. Enhancing library diversity, refining affinity maturation, and addressing the complexity of disease targets are key areas for improvement. Solutions include creating more diverse phage libraries [267], optimizing selection methods [260], and integrating advanced screening techniques to target complex biological environments [262].

In conclusion, phage display has become a cornerstone of drug discovery with its ability to identify and optimize therapeutic molecules. Ongoing advancements promise to further enhance the discovery and development of innovative treatments, positioning phage display as a critical tool in future therapeutic innovation.

In conclusion, phage-display technology has profoundly impacted drug discovery, enabling the rapid identification and optimization of therapeutic molecules. From its fundamental principles and diverse applications to recent technological advancements, this technology continues to evolve, offering innovative solutions for target identification, lead optimization, antibody engineering, and peptide drug development. However, challenges such as enhancing library diversity, improving affinity maturation, and addressing biological complexity remain. Addressing these issues will be crucial for maximizing the potential of phage display in developing new treatments for various diseases. As the field advances, ongoing research and technological innovation will further enhance the precision and efficiency of drug-discovery processes, promising a future where phage display plays an even more significant role in therapeutic development.

## 10. Safety and Regulatory Considerations

As the field of phage-based therapeutics continues to advance, it brings with it unique challenges and considerations that must be addressed to ensure the safe and effective translation of these innovative treatments into clinical practice. This section of the manuscript delves into the critical aspects of safety and regulatory frameworks that are essential for the development and approval of phage therapies. It highlights the complexities of host–phage interactions, the potential for off-target effects, and the dynamic nature of phage genomes, all of which underscore the need for rigorous safety evaluations. Additionally, the section outlines the regulatory pathways in the United States and Europe, emphasizing the importance of harmonizing global standards to streamline the approval process and bring phage-based solutions to patients in need.

### 10.1. Safety Considerations

Host Interactions: Phages interact with host organisms in complex ways, potentially eliciting immune responses or causing adverse effects. These interactions may vary depending on factors such as the phage’s lifecycle (lytic vs. lysogenic) and the patient’s immune status. For instance, Sarker et al. highlighted the potential of phages to elicit immune responses, as seen in their study on the safety of oral phage therapy in children from Bangladesh, indicating the need for a comprehensive evaluation to ensure a favorable safety profile [103]. Furthermore, repeated exposure to phages may lead to the development of neutralizing antibodies, which could reduce the efficacy of phage therapy over time.

Understanding host–phage interactions is crucial for predicting and mitigating potential immunogenicity or adverse inflammatory responses. Additionally, some studies have shown that repeated exposure to phages may lead to the development of neutralizing antibodies, which could reduce the efficacy of phage therapy over time [95,268]. A comprehensive evaluation of these interactions is essential for ensuring a favorable safety profile and minimizing risks associated with phage therapy.

Off-Target Effects: Phage-based therapeutics may inadvertently target non-pathogenic or beneficial bacteria, disrupting the microbiome and causing unintended consequences. The human microbiome plays a vital role in maintaining health, and its disruption can lead to dysbiosis, potentially resulting in conditions such as gastrointestinal disorders or increased susceptibility to infections. Assessing off-target effects, including potential impacts on the microbiome, is critical for ensuring the safety and efficacy of phage treatments. Advanced bioinformatic tools and in vitro assays are increasingly being used to predict and evaluate these effects prior to clinical application [103,269].

Genomic Stability: Phage genomes are inherently dynamic, capable of undergoing mutations or recombination events that can alter their characteristics, such as virulence, host range, or resistance to bacterial defenses. This genomic plasticity raises concerns about the long-term safety and predictability of phage therapy. Monitoring genomic stability is crucial for detecting any changes that may impact the safety or efficacy of phage therapeutics. Additionally, the potential for phage genomes to integrate into the bacteria host’s genetic material, particularly in the case of lysogenic phages, necessitates careful evaluation to avoid unintended genetic modifications that could contribute to bacterial resistance or other adverse outcomes [32,270].

### 10.2. Regulatory Frameworks

The advancement of phage-based therapeutics represents a promising frontier in medicine, offering innovative solutions for a range of challenging health conditions. However, navigating the regulatory landscape is crucial to ensuring that these therapies are both safe and effective. A recent review by Yang et al. (2023) discussed details about the regulations regarding phage therapy in many countries [271]. This section delves into the regulatory frameworks governing phage-based therapeutics in key regions, including the United States and Europe, as well as ongoing efforts towards international harmonization. By exploring the guidelines set forth by the FDA and EMA, and examining global initiatives for standardization, this discussion highlights the essential role of regulatory oversight in the development and approval of phage-based products. Understanding these frameworks is vital for researchers and developers aiming to bring these cutting-edge therapies to market and ultimately improve patient outcomes.

FDA Regulation: In the United States, phage-based therapeutics are regulated by the Food and Drug Administration (FDA) under the same guidelines as other biologics or antimicrobial agents. Developers must adhere to stringent regulatory requirements, including preclinical testing to demonstrate safety and efficacy, as well as rigorous clinical trials to evaluate the therapeutic potential in human patients. The FDA’s regulatory framework is designed to ensure that phage-based products meet high standards of quality, safety, and efficacy before they can be approved for commercial use. The FDA also provides pathways for expedited review and approval of phage therapies in cases of unmet medical need or when treating life-threatening conditions [272,273].

EMA Regulation: In Europe, the European Medicines Agency (EMA) oversees the regulation of phage-based therapeutics through the centralized marketing authorization procedure [274]. This process involves a thorough review of the product’s quality, safety, and efficacy by EMA’s Committee for Medicinal Products for Human Use (CHMP). The EMA requires developers to submit comprehensive data from preclinical and clinical studies, as well as detailed manufacturing information, to ensure that phage-based therapeutics are safe and effective for their intended use. The EMA also encourages early dialogue with developers to address any regulatory challenges and facilitate a smoother approval process [275,276].

International Harmonization: Efforts are underway to harmonize regulatory standards for phage-based therapeutics globally, which would facilitate the development, approval, and commercialization of these products across different regions [271]. International organizations, such as the International Council for Harmonisation of Technical Requirements for Pharmaceuticals for Human Use (ICH), are working to create common guidelines and standards for the evaluation of phage-based therapies. Harmonization of regulatory frameworks is expected to reduce the burden on developers, streamline the approval process, and ensure that phage therapies meet consistent safety and efficacy standards worldwide [277].

In conclusion, safety and regulatory considerations are integral aspects of the development and approval process for phage-based therapeutics. By addressing key safety concerns, adhering to regulatory requirements, and advancing international harmonization efforts, researchers and developers can accelerate the clinical translation and commercialization of phage-based products, ultimately benefiting patients and public health.

## 11. Challenges and Future Directions

Despite significant progress in phage-based therapies, several challenges remain that must be addressed to fully realize their potential. This section discusses key challenges in the field of phage therapy and outlines future directions for research and development.

### 11.1. Standardization and Regulatory Considerations

Establishing standardized protocols for phage production, safety, and efficacy is critical for regulatory approval and clinical translation. Harmonizing methodologies and endpoints can streamline the regulatory process and ensure product consistency [274]. Recent advancements in automated phage production systems are promising as they offer robust and scalable methods for consistent production [278] [1]. Improvements in quality control and regulatory frameworks are essential for maintaining high standards across phage products [279].

Designing well-controlled clinical trials with appropriate endpoints and patient populations is crucial for generating robust data to support regulatory submissions. Addressing challenges such as patient recruitment, trial duration, and endpoint selection can enhance the likelihood of regulatory approval [280]. Additionally, post-market surveillance strategies must be implemented to monitor the safety and effectiveness of approved phage-based products, contributing to the ongoing assessment of product safety profiles [281].

The regulatory landscape for phage therapy is evolving, with efforts to establish clearer guidelines and approval pathways. Collaboration between regulators, researchers, and industry stakeholders is crucial for advancing regulatory frameworks and facilitating the translation of phage therapies from the laboratory to the clinic [272].

### 11.2. Therapeutic and Technological Challenges

The host immune response poses a significant challenge to the success of phage therapy. Neutralizing antibodies can limit the efficacy of phage treatment by clearing phages from the bloodstream and tissues. Strategies to evade or modulate the host immune response are needed to enhance the effectiveness of phage therapy [278]. Understanding the pharmacokinetics of phages in vivo is also essential for optimizing treatment regimens and dosing strategies [279,282].

Bacterial resistance to phages is another critical concern. Bacteria can develop mechanisms to evade phage infection, such as surface receptor mutations, CRISPR-Cas systems, and biofilm formation. Strategies to overcome bacterial resistance, such as phage cocktails, genetic engineering, and combination therapies, are crucial for maintaining treatment efficacy [283].

Engineering phages with desired properties, such as enhanced infectivity, stability, and targeting specificity, presents technical challenges. Methods for phage genome manipulation, capsid modification, and ligand display require further optimization [4]. Additionally, developing effective delivery strategies to target phages to specific infection sites or tissues remains a challenge. Advances in nanoparticle-based delivery systems, bioengineering approaches, and targeted formulations hold promise for overcoming these barriers [68].

### 11.3. Future Directions

Combination Therapies: Exploring combination therapies that synergistically target bacteria using phages along with antibiotics, bacteriocins, or immune-modulating agents holds promise for overcoming resistance mechanisms and improving treatment outcomes [274]. Research indicates that phage–antibiotic combinations can significantly improve clinical outcomes in multidrug-resistant infections [281].

Personalized Medicine: Embracing personalized medicine approaches that tailor phage therapy to individual patient characteristics represents a future direction for optimizing treatment outcomes. Precision medicine strategies can enhance treatment efficacy, minimize adverse effects, and improve patient outcomes [280].

Biotechnology Innovation: Leveraging advances in biotechnology, such as synthetic biology, genome editing, and high-throughput screening, offers new opportunities for phage therapy development. Engineering designer phages with custom-designed properties, developing novel delivery platforms, and screening large phage libraries for therapeutic candidates are areas ripe for innovation [281].

Global Accessibility: Ensuring global access to phage therapy, particularly in low-resource settings, is a significant challenge. Efforts to develop cost-effective phage products and establish infrastructure for production and distribution are crucial. Expanding initiatives like those by the Phage Therapy Center in Tbilisi, Georgia, and fostering international collaboration will be key to addressing global health challenges related to antibiotic resistance [284].

In conclusion, addressing the challenges and exploring future directions outlined in this section will be instrumental in advancing the field of phage therapy and realizing its full potential as a transformative approach for combating antibiotic-resistant infections and other diseases.

## 12. Conclusions

Phage-based drug development represents a transformative frontier in modern medicine, extending far beyond traditional phage therapy for bacterial infections. This approach harnesses the versatility of bacteriophages for a wide range of applications, including cancer treatment, vaccine development, and drug-delivery systems (DDS). By engineering phages to target specific disease markers, deliver therapeutic agents, or stimulate immune responses, researchers are uncovering novel strategies to address complex medical challenges, such as enhancing therapeutic efficacy, combating a diverse range of pathogens, and overcoming traditional drug-delivery barriers.

In cancer treatment, phages are being engineered to selectively target and destroy tumor cells or deliver cytotoxic agents directly to cancerous tissues, enhancing therapeutic efficacy while minimizing collateral damage to healthy cells. This precision targeting promises to revolutionize oncological care by offering more personalized and effective treatments. Similarly, phage-based vaccines are emerging as innovative solutions for immunization, leveraging phages’ ability to present antigens in a way that elicits strong and specific immune responses. These vaccines hold the potential for combating a wide array of pathogens and diseases.

The development of phage-based drug-delivery systems (DDS) is another promising avenue. Phages can be designed to deliver drugs directly to specific cells or tissues, overcoming traditional delivery barriers and improving the precision of treatments. This capability is particularly valuable in targeting diseases with complex biological contexts or where conventional delivery methods fall short.

Despite these exciting advancements, several challenges must be addressed. Ensuring the safety and efficacy of phage-based therapies requires rigorous evaluation of host–phage interactions, potential off-target effects, and genomic stability. Regulatory pathways must adapt to these novel applications, necessitating clear guidelines and standardized protocols to facilitate approval and commercialization. Additionally, optimizing phage engineering techniques and delivery systems remains an ongoing challenge, as does overcoming biological barriers and resistance mechanisms.

Overall, phage-based drug development holds immense promise for advancing therapeutic strategies across diverse medical fields. Continued research, innovation, and collaboration are essential to fully realize the potential of phages in addressing both bacterial infections and other complex diseases. As the field evolves, it offers the opportunity to transform how we approach treatment, paving the way for more effective, personalized, and targeted medical solutions.

## Figures and Tables

**Figure 1 antibiotics-13-00870-f001:**
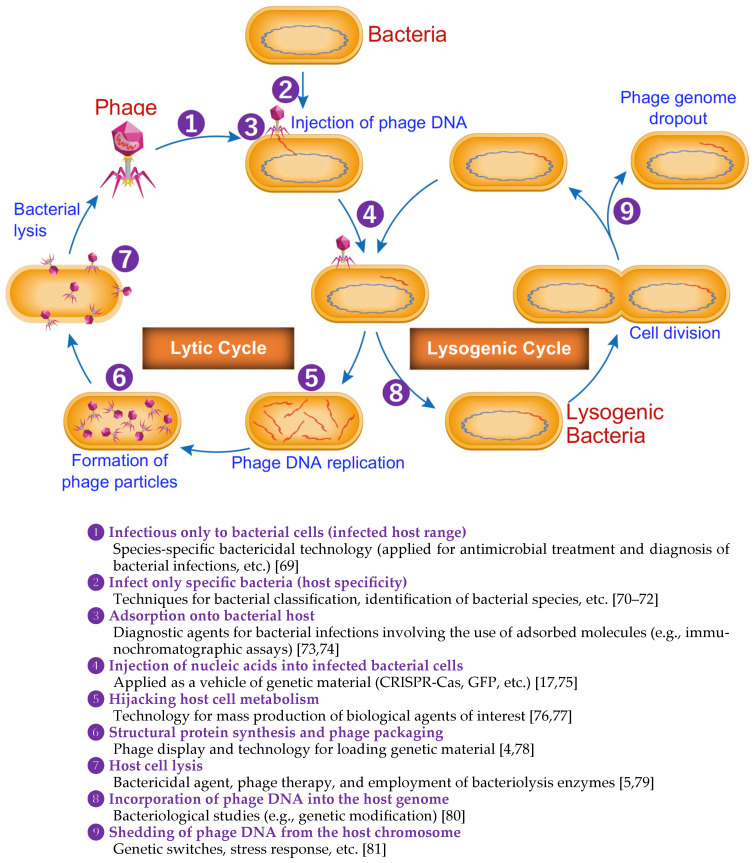
Life cycle of bacteriophages and its application to medicine [4,5,17,69,70,71,72,73,74,75,76,77,78,79,80,81]. Phages can be divided into two main groups based on their life cycles: lytic phages, which definitely destroy bacterial host upon infections, and lysogenic phages, which stay dormant in bacteria and replicate their DNA without destroying the host until they are induced to enter lytic life cycle [82]. Life cycles of phages have been adopted for use in various ways in medicine, industry, and research.

**Figure 2 antibiotics-13-00870-f002:**
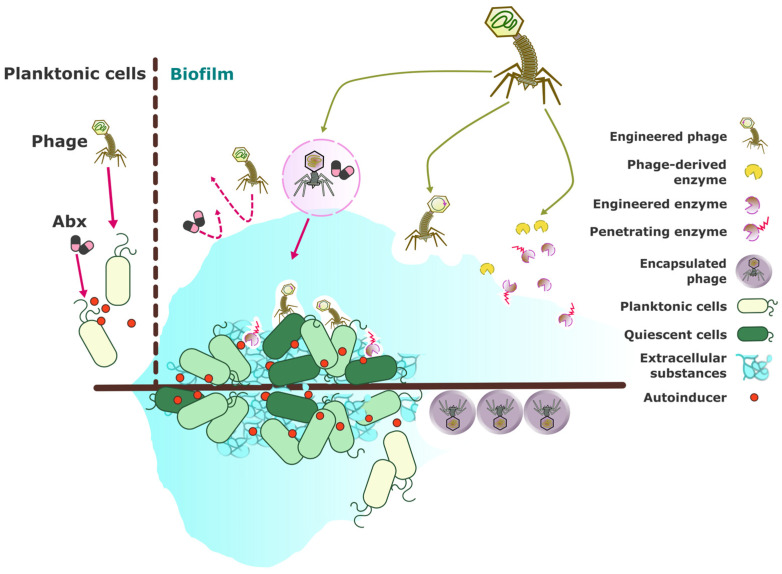
Schematic diagram of phage-based treatments for biofilm-forming bacteria. Biofilms provide a protective environment for pathogenic bacteria, often rendering conventional antimicrobial treatments that target planktonic bacteria ineffective. Strategies utilizing phages and phage-derived products to overcome biofilm-associated infections include (1) combinative phage-antibiotic or phage-antibiofilm regimens [141,142,143,144]; (2) engineering phages and phage-derived products to enhance the targeting of biofilm-associated bacteria [145,146,147,148] and induce biofilm dispersal [146,149,150]; and (3) formulation and encapsulation strategies for the effective delivery of phages to biofilm sites [98,151,152]. Abx: antibiotic.

**Figure 3 antibiotics-13-00870-f003:**
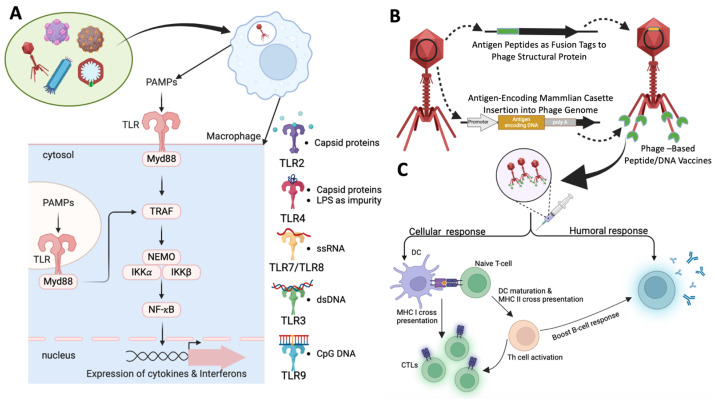
Mechanism of induction of immune responses by phage-based vaccines. (**A**) Mechanism of self-adjuvanting effect of phages and how phage prime innate immune response: Wide variety of phages, filamentous, tailed, or icosahedral were utilized as vaccine vectors. The intrinsic components of the phages such as the repetitive ordered capsid proteins, DNA, RNA, and CpG Islands can act as pathogen-associated molecular patterns (PAMPs) and can bind either cell-surface or endosomal Toll-like receptors (TLRs) such as TLR 2, 3, 4, 7, 8, and 9. These receptors are pattern-recognition receptors (PRRs) that are primarily seen in innate immune cells and functions in pathogen identification. The viral structural proteins are known to bind TLR2 and TLR4, whereas the DNA, RNA, and CpG Islands bind TLR3, TLR7 and 8, and TLR 9, respectively [181,182]. In addition to phage components, impurities derived from lysates can also induce an immune response, ex. LPS. The PAMPs’ post-binding of respective TLRs can activate Myd88 pathway and downstream signalling, leading to phosphorylation of IKK complex, IKK-α, IKK-β, and NEMO subunits. The IKK complex when phosphorylated frees NF-κB enables nuclear translocation that furthers the expression of array of pro-inflammatory cytokines and interferons imparting adjuvant-like effect [182]. (**B**) Construction of peptide vaccine and DNA vaccine using phages: Peptide vaccines were prepared by inserting antigenic epitopes as fusion tags to the structural proteins of the capsids. This technique enables the display of the antigens on the surface that could mediate an antigen-specific immune response. In case of DNA vaccine, the phage genome is inserted with an antigen-encoding gene cassette. In this way, the DNA of the antigen is encapsulated in phage head and is delivered to the target immune cells wherein the DNA is transcribed and translated to express antigen of interest [183]. (**C**) Mechanism of phage vaccine imparting antigen-specific response: The peptide epitope (from peptide vaccine) or the antigens expressed (from DNA vaccine) can impart an antigen-specific immune response. The phage vaccines taken up by APCs were processed and the antigens were presented to naive T-cells via MHC II or MHC I. This process of the presentation activates the naïve T-cells to become CTLs or Th cells. The activated Th cells further boost the memory CTL production and also impart a boost towards antigen-specific antibody production via humoral B-cell responses. The adaptive immune response is known to be further boosted by the self-adjuvanting activity of the phages itself via production of wide array of pro-inflammatory cytokines [184]. Created with templates from BioRender.com.

**Table 1 antibiotics-13-00870-t001:** Comprehensive overview of phage-facilitated medical and technological advancements.

Phage-Enabled Techniques and Their Applications in Modern Medicine and Biotechnology	Relevant Strategies and Methodologies	References
1. Gene-Targeted Bacterial Killing	Loading CRISPR-Cas13 into the capsid Loading CRISPR-Cas9 into the capsid Loading crRNA of Cas3 into the capsid	[17,18,19,20,21]
2. Delivery of Antimicrobial Genes	Loading toxin genes into the capsid Loading antimicrobial peptides into the capsid	[22,23,24,25]
3. Phage-Mediated Antimicrobial Agent Delivery	Displaying antibodies against target bacteria on the phage surface Conjugating nanoparticles to the capsid surface of phages binding to target bacteria	[26,27,28,29]
4. Strict Lytic Cycle Maintenance	Deleting lysogenic genes	[30]
5. Inhibition of Biofilm Formation	Loading enzymes that degrade biofilms into the phage Utilizing depolymerase of phages	[31,32,33]
6. Modification of Phage Host Range	Modifying tail fibers Introducing random mutations into the host recognition region of tail fibers	[34,35,36,37,38,39]
7. Strategies to Overcome Phage Resistance	Introducing factors that inhibit bacterial defense systems	[24,40]
8. In Vivo Phage Stabilization	Introducing mutations into capsid genes PEGylation of the capsid Adding peptides to the capsid Improving stability in temperature and pH	[41,42,43,44,45]
9. Enhancing Antibiotic Sensitivity	Loading genes that inhibit resistance into the phage Using CRISPR-Cas to cleave resistance genes Knocking down resistance genes (e.g., sRNA)	[46,47,48,49]
10. Endotoxin Shock Suppression	Deleting the lytic enzyme of the phage	[50,51]
11. Phage-Based Vaccines	Displaying antigens on the phage surface (targets: Yersinia, Neisseria, HIV, HPV, Influenza) Inserting pathogen DNA into the capsid (DNA vaccine) (targets: HBV, HSV, Chlamydia)	[52,53,54,55,56,57,58]
12. Phage in Gene Therapy Applications	Creating a hybrid of phage and adenovirus	[59,60]
13. Phage-Mediated Virus Suppression	Displaying virus receptors on the phage	[61]
14. Phage for Diagnostic Applications	Introducing reporter genes	[62]
15. Phage-Assisted Bone Regeneration	Promoting bone regeneration using RGD-modified M13 phage	[63,64]
16. Phage-Assisted Skin Regeneration	Creating phage films for skin regeneration	[65]
17. Phage-Assisted Nerve Regeneration	Utilizing nano-micro hierarchical structures with M13 phage for nerve regeneration	[66,67]

**Table 2 antibiotics-13-00870-t002:** Development stage of clinical studies using bacteriophages registered on ClinicalTrials.gov on 15 August 2024 *.

Status	Completed or Terminated						On Going or Not Yet Recruiting			
Phase	Pre-Clinical or Case Study	Phase					Pre-Clinical or Case Study	Phase		
		I	I/II	II	II/III	III		I/II	II	III
*Staphylococcus aureus*	1	2	1			1	3	3	3	
Coagulase-negative Staphylococci	1						3			
*Pseudomonas aeruginosa*	2	1	4	1		1	1	3	1	
*Escherichia coli*	2	3	1		1	1	2	2	1	1
*Enterococcus* spp.	1				1	1	2	1		
*Klebsiella*	1	1	1			1	1	2		
*Acinetobacter baumannii*	1		1				1		1	
*Streptococcus*					1	1				
*Proteus*	1					1	1			
Non-tuberculosis *Mycobacteria*	1						1			
*Achromobacter xylosoxidans*	1						1			
*Stenotrophomonas maltophilia*	1						1			
*Bacteroides fragilis*	1						1			
*Shigella*			1							

* Note: The numbers in the table represent the number of research projects (clinical trials). In some cases, phage cocktails targeting different bacteria are included, with each counted as one project. Most of the research projects are currently ongoing, so many have not yet been published. The table was compiled by the authors using the ClinicalTrials.gov database from the National Library of Medicine (https://clinicaltrials.gov/, accessed on 15 August 2024), and there are no comprehensive references for the entire table. It provides an overview of the current status of clinical studies using bacteriophages.

**Table 3 antibiotics-13-00870-t003:** Comparative analysis of conventional and synthetic approaches to overcome challenges in phage therapy.

	Challenges in Phage Therapy	Conventional Approach	Synthetic Approach
1	Narrow host range	The use of phage cocktail [111,112]	Genetic manipulation of receptor-binding protein [30,34]
2	Emergence of phage-resistant bacteria	Phage cocktail; combined therapy of antibiotic and phage [113,114]	Genetic manipulation of receptor-binding protein [115]; incorporation of small RNAs or CRISPR-Cas system to silence antibiotic resistance determinant [17,49] or delivery of genes encoding proteins to sensitize bacteria against antibiotics [116]
3	Necessitate identification of phage with therapeutic effect against patients’ isolates (personalized medicine)	Establish phage biobanks (isolating large phage collections) [117]	Engineering of phage tail fibers to alter host range [35,36,38]
4	Rapid clearance by reticuloendothelial system (RES)	Multiple phage dosing [118]	Phage capsid protein mutagenesis [41]; PEGylation of phage particles [43]
5	Phage pharmacokinetics (bioavailability through oral administration)	Pharmacological neutralization of gastric acid [119]	Encapsulation of phage in nanoparticles [120,121]
6	Limited accessibility to biofilm-producing bacteria	Use only phages with intrinsic biofilm-degrading properties [122,123], or combined therapy using phage and biofilm-degrading enzymes [124]	Engineered phages expressing biofilm-degrading enzymes [31]
7	Difficulties in defining pharmacokinetics (e.g., MIC)	Standardize routes and dosages of administration (required specified combinations of phage-host for each infection) [125]	Generation of non-proliferative anti-bacterial phage capsids [17]
8	Safety concern: risk of horizontal gene transfer	The use of phage-derived endolysin [126]	Development of well-characterized, non-propagating phages [127], introduction of antibacterial cargo using phagemids [128,129] or phage-inducible chromosomal islands (PICIs) [20]
9	Presence of potential hazardous genes in phage genome (toxin, virulence, antibiotic resistance genes, etc.)	Obligate virulent phage is preferred in therapy [113]; whole-genome analysis should be done in the first place	Custom-made phage can be generated easily using current techniques [30,34,130]; the use of self-destructive engineered phage (conjugation to gold nanorods) [29]
10	Low purity and potential toxin contamination in phage preparation	Purification by CsCl density gradient and ion exchange column [113] or affinity chromatography [131]	The use of cell-free system (cell-free-transcription-translation, TXTL) for phage production [130]

**Table 4 antibiotics-13-00870-t004:** Reports on the synergistic effects of combined phage and antibiotic therapy.

	Study	Target Bacteria	Phage (Dosages)	Antibiotic (Dosages)	References
1	Racenis et al., 2023	Multidrug-resistant *P. aeruginosa*	Phages PNM and PT07 (Titer of 10^7^ PFU/mL)	Ceftazidime/Avibactam (2.5 g) and Amikacin (750 mg)	[133]
2	Kebriaei et al., 2023	Methicillin-resistant *S. aureus* (MRSA) strains and their daptomycin-nonsusceptible vancomycin-intermediate (DNS-VISA)	Phages Intesti13, Sb-1, and Romulus (10^7^ PFU/well)	Daptomycin, vancomycin, and ceftaroline at 0.5× MBIC or 1× MBIC	[105]
3	Altamirano et al., 2022	*A. baumannii* AB900	Phages øFG02 (range: 10^2^–10^8^ PFU/mL)	Ceftazidime (range: 1–512 mg/mL)	[135]
4	Cano et al., 2021	*K. pneumoniae* complex KpJH46	Phage KpJH46Φ2(6.3 × 10^10^ phages in 50 mL for a total of 40 doses)	Minocycline, 100 mg	[136]
5	Morales et al., 2020	*S. aureus*	Three Myoviridae bacteriophages AB-SA01 (109 PFU/mL)	lucloxacillin, Cefazolin, Vancomycin, Ciprofloxacin, Rifampicin	[137]
6	Jault et al., 2019	*P. aeruginosa*	cocktail of 12 natural lytic anti-*P. aeruginosa* bacteriophages (PP1131; 1 × 10^6^ PFU]/mL)	1% sulfadiazine silver emulsion cream	[138]
7	Osman et al., 2023	MDR *A. baumannii*	Multiple phage cocktails (C2P24, AC4, C2P21, and C1P12)	Minocycline	[139]

**Table 5 antibiotics-13-00870-t005:** List of phages utilized for cancer theragnostic.

Phage	Functional Peptide Display/Cargo	Tumor Type	Mode of Therapy	Preclinical Model	Therapy Outcome	References
M13	WDC-2 phage displaying melanoma cell targeting peptide TRTKLPRLHLQS	Melanoma	Immunomodulatory	Subcutaneous B16-F10 tumor model in mice	Delayed tumor growth and increased survival	[197]
λ Phage	Display of human ASPH-derived proteins	Hepatocellular carcinoma	Immunotherapy—Delivery of antigen for vaccine effect	Prophylactic vaccination schedule in BNL HCC subcutaneous model	Prophylactic and therapeutic immunization significantly delayed HCC growth and progression	[201]
Hybrid M13/AAV	RGD4C peptide CDCRGDCFC that binds to α_v_β_3_ integrin cell surface receptor on Glioblastoma	Glioblastoma	Gene therapy—Grp78 expression	Intracranial implantation of U87 glioblastoma cells	Suppressed the growth of orthotopic glioblastoma	[209]
M13 phage	*Fusobacterium nucleatum* binding M13 phages	Colorectal cancer	Immunomodulatory	Orthotopic CT26 murine model	Precise scavenging of pro-tumor bacteria of *Fusobacterium nucleatum*, thereby blocking immunosuppressive myeloid-derived suppressor cells augmentation in the tumor microenvironment.	[28]
2nd generation M13 vector	CDCRGDCFC (RGD4C) ligand that binds to α_v_β_3_ integrin	Chondrosarcoma	Gene therapy–tumor necrosis factor-related apoptosis-inducing ligand (TRAIL) expression	Subcutaneous implantation of SW1353-GFP-Luc cells	Decreased tumor size with nil side effects	[210]
TPA (transmorphic phage/AAV)	Tumour targeting ligand, CDCRGDCFC (RGD4C)	Hepatocellular carcinoma	Therapeutic gene cassette that expresses TRAIL	N/A	Selective and efficient delivery of the *tmTRAIL* gene to HCC cells that induced apoptotic death of HCC cells	[211]
Transmorphic phage/AAV, TPA	Double-cyclic tumour-targeting ligand, RGD4C ligand	Medulloblastoma	Delivering transgene expressing the tumor necrosis factor-alpha (TNFα)	Subcutaneous Daoy medulloblastoma xenograft mice model	Selective tumor homing, targeted tumor expression of TNFα, apoptosis, and destruction of the tumor vasculature	[212]
M13/AAV	RGD4C ligand on the pIII minor coat protein for targeted therapy Histidine-rich endosomal escape peptide, H5WYG	Chondrosarcoma	Delivery of TNFα transgene	Subcutaneously established SW1353 xenograft in athymic mice	Complete elimination of tumor growth and eradication of the tumor size and tumor viability	[213]
M13 bacteriophage	Chemical cross-linking and biomineralization of palladium nanoparticles	Breast cancer	Delivery of palladium nanoparticle for photothermal therapy and NLG919, a nontoxic IDO1-selective inhibitor	Subcutaneous breast cancer model using 4T1 cells	Induced immunogenic death of tumor cells with down-regulated IDO1 expression	[214]
M13	Fn-binding phages	Colon carcinoma	Immunomodulatory and reversing chemoresistance	Caecum implantation of CT26 cells in BALB/c mice	Modulated gut microbiota to augment chemotherapeutic effect	[215]
M13	Peptide (SYPIPDT) that is able to bind the epidermal growth factor receptor (EGFR) Chemical conjugation of Rose Bengal (RB) photosensitizing molecules on the capsid surface	Epidermoid carcinoma	Photodynamic therapy	N/A	M13_EGFR_–RB derivatives generated intracellular reactive oxygen species activated by an ultralow intensity white light irradiation, thereby killing the cancer cells	[216]
T7	Cancer homing peptide pep42 (CTVALPGGYVRVC) targeting the grp78 on cancer cells	Melanoma	Mammalian expression cassette of the cytokine granulocyte macrophage-colony stimulating factor (GM-CSF)	Subcutaneous B16F10 xenografts	Inhibited tumor growth by 72% compared to the untreated control.	[217]
M13	Engineered to display the EC and TM domains of human HER2 (ECTM phages) or its splice variant Δ16HER2	Breast carcinoma	Immunotherapy-Delivery of antigen for vaccine effect	Δ16HER2-expressing epithelial tumor cell lines mice	Anti-HER2 vaccination induced a significant anti-HER2 antibody response and controls tumor growth.	[218]
M13	Display of anti-CD40 DARPin into the gene of the pIII coat protein for CD40 targeting	Colon adenocarcinoma	Immunotherapy—In situ vaccines	Subcutaneous MC38 xenografts	Significant accumulation of the phages and activation of DCs at the tumor site, reversing the immunosuppressive tumor microenvironment	[219]
λ Phage	Tumor selectivity of the cargo, apoptin	Breast Carcinoma	Gene therapy	BT-474 cells subcutaneous xenograft	Implanted BT-474 human breast tumor successfully responded to the systemic and local injection of untargeted recombinant λ NBPs	[220]
λ Phage	Display of displaying a HER2/neu derived peptide GP2	Breast carcinoma	Immunotherapy—Delivery of antigen for vaccine effect	Subcutaneous TUBO cell implant	Robust CTL response against HER2/neu-positive tumor challenge in both prophylactic and therapeutic settings	[221]
T4-AAV	RGD peptide (CDCRGDCFC), a cell surface targeting ligand, when fused to the tip of Hoc fiber	HEK293T	Gene delivery, Protein Delivery & Genome editing	N/A	Delivered full-length dystrophin gene and performed genome editing, gene recombination, gene replacement, gene expression, and gene silencing.	[222]
T4	Display of Catalase protein on phage heads Chemically coupled chlorin e6 (Ce6), a photosensitizer	Breast cancer	Photodynamic therapy	Subcutaneous 4T1 cancer cell model	Relieved tumor hypoxia and enabled Ce6 to produce ROS for effective tumor inhibition	[223]
T7 phage	Display of neoepitopes derived from mutated proteins of melanoma tumor cells	Melanoma	Immunotherapy—Delivery of neoepitopes for vaccine effect	Subcutaneous B16F10 xenografts	Rapid production of vaccines that can deliver mutated peptides and stimulate an appropriate B cell response	[224]

## Data Availability

Not applicable.

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
