# Peer review of "A Comprehensive Review on Phage Therapy and Phage-Based Drug Development"

_antibiotics, 2024, doi:10.3390/antibiotics13090870_

Round 1

Reviewer 1 Report

Comments and Suggestions for Authors

Cui et al showed a very comprehensive review about phage utilization in different aspects including MDR until cancer treatment. I found this is fascinating because they can compile hundreds references. Unfortunately, there are couple major and minor concerns as I stated below:

Major point:

1. The review is very superficial. They take one source, and just very minorly describe what are the findings. For example:

Page 19, line 738-739, "Carroll-Portillo and Lin (2019) further describe how phages can influence innate immune signaling pathway".

For me as reader, I do not get any valuable information.

Please rewrite not only what I mentioned but also other parts.

2. It is important for the authors to declare whether they used AI help or not.

Several minor points:

1. Line 284, Enterococcus faecalis must be in Italic

Author Response

Reviewer-1

Comments and Suggestions for Authors

Cui et al showed a very comprehensive review about phage utilization in different aspects including MDR until cancer treatment. I found this is fascinating because they can compile hundreds references. Unfortunately, there are couple major and minor concerns as I stated below:

Major point:

  1. The review is very superficial. They take one source, and just very minorly describe what are the findings. For example:

Page 19, line 738-739, "Carroll-Portillo and Lin (2019) further describe how phages can influence innate immune signaling pathway".

For me as reader, I do not get any valuable information.

Please rewrite not only what I mentioned but also other parts.

  1. It is important for the authors to declare whether they used AI help or not.

Several minor points:

  1. Line 284, Enterococcus faecalis must be in Italic

Response to Reviewer-1

Dear Reviewer

Thank you for your thoughtful and constructive comments on our manuscript titled “A Comprehensive Review on Phage Therapy and Phage-Based Drug Development”. We greatly appreciate your time and effort in reviewing our work. We have carefully considered each of your points and have revised the manuscript accordingly. Below, we provide detailed responses to your comments and outline the changes made in the revised version.

Major Points:

  1. The review is very superficial. They take one source, and just very minorly describe what the findings are. For example, on Page 19, line 738-739, “Carroll-Portillo and Lin (2019) further describe how phages can influence the innate immune signaling pathway.” For me as a reader, I do not get any valuable information. Please rewrite not only what I mentioned but also other parts.

Response:

Thank you for this valuable feedback. We agree that more detailed explanations are needed to enhance the depth and clarity of the discussion, especially on key topics like phages’ influence on the innate immune signaling pathway. We have rewritten the relevant sections, including the example you mentioned, by providing a more thorough explanation of the findings. For instance, we expanded on how phages interact with toll-like receptor (TLR) and the downstream effects on cytokine production, which plays a critical role in immune modulation. Similar revisions have been applied to other parts of the manuscript to provide greater insight into the reviewed literature.

  1. It is important for the authors to declare whether they used AI help or not.

Response:

We have already included a statement in the “Declaration of Generative AI Use” section at the time of submission, stating: “We confirm that AI was utilized in the development of this manuscript for language refinement and ensuring coherence. The intellectual content, analysis, and interpretations remain entirely the work of the authors.” This declaration was made in accordance with the journal’s guidelines to ensure full transparency.

Minor Points:

  1. Line 284, Enterococcus faecalis must be in Italic.

Response:

Thank you for pointing this out. We have corrected the formatting of Enterococcus faecalis and reviewed the manuscript to ensure all species names are correctly italicized.

We hope that the revisions meet your expectations, and we believe they have strengthened the manuscript.

Reviewer 2 Report

Comments and Suggestions for Authors

The authors aimed to write a comprehensive review of phage therapy, highlighting the uses of phages in anti-bacterial/biofilm and -cancer therapy, vaccine development and gene delivery. The review also discusses the challenges associated with phage therapy such as their stability, immune responses, the complications in regulatory approval, and the ability of bacterial hosts to gain resistance to respective phages. In addition to the direct application of bacteriophages in therapies, authors also noted other ways in which they can be used – for instance, in the delivery of genes and antimicrobial agents, evolution of vaccine candidates and their multivalent display for stronger immune responses etc.

The authors have chosen an excellent topic for the review. The review is comprehensive and organized. It provides a thorough overview of the progress made by the scientific and medical communities in these areas with up-to-date information from recently published articles. Overall text is very well structured into sub-sections, and the figures and the tables communicate the message very well. This review can be a good starting point for young researchers and non-experts willing to get a quick overview of the topic.

                Although the review broadly covers the topic, many changes can be made to it in order to attract wide audience (experts and senior researchers included). For the reasons discussed below, I believe that a few minor changes could be made before its publication to make it a highly cited, leading review on the topic. I hope the authors find these suggestions considerable and incorporate them in the final manuscript.

1)      The review is very lengthy. This can be shortened by eliminating general/redundant statements, which can be found at several places in the review. For example, the lines 239-240, 414-416, 446-448, 553-554 etc. have almost the same meaning. Moreover, removing those statements at most places will not change the way the message is communicated.

2)      After the word-count is cut down significantly, it would be great if the authors can make an attempt to critically analyze / discuss the work they have cited. For ex, in lines 427-429 the authors write that Zuo et al develop a biofilm-responsive coat – it would be great to discuss what this coat material is composed of, and what component acts as the trigger that would release the phages specifically at biofilms. Similarly, what are the techniques used to engineer bacteriophages. Such topics may be explained in a couple of sentences, which when done at all the places in the review will make it even more interesting for the readers.

Author Response

Reviewer-ï¼’

Comments and Suggestions for Authors

The authors aimed to write a comprehensive review of phage therapy, highlighting the uses of phages in anti-bacterial/biofilm and -cancer therapy, vaccine development and gene delivery. The review also discusses the challenges associated with phage therapy such as their stability, immune responses, the complications in regulatory approval, and the ability of bacterial hosts to gain resistance to respective phages. In addition to the direct application of bacteriophages in therapies, authors also noted other ways in which they can be used – for instance, in the delivery of genes and antimicrobial agents, evolution of vaccine candidates and their multivalent display for stronger immune responses etc.

The authors have chosen an excellent topic for the review. The review is comprehensive and organized. It provides a thorough overview of the progress made by the scientific and medical communities in these areas with up-to-date information from recently published articles. Overall text is very well structured into sub-sections, and the figures and the tables communicate the message very well. This review can be a good starting point for young researchers and non-experts willing to get a quick overview of the topic.

Although the review broadly covers the topic, many changes can be made to it in order to attract wide audience (experts and senior researchers included). For the reasons discussed below, I believe that a few minor changes could be made before its publication to make it a highly cited, leading review on the topic. I hope the authors find these suggestions considerable and incorporate them in the final manuscript.

1) The review is very lengthy. This can be shortened by eliminating general/redundant statements, which can be found at several places in the review. For example, the lines 239-240, 414-416, 446-448, 553-554 etc. have almost the same meaning. Moreover, removing those statements at most places will not change the way the message is communicated.

2) After the word-count is cut down significantly, it would be great if the authors can make an attempt to critically analyze / discuss the work they have cited. For ex, in lines 427-429 the authors write that Zuo et al develop a biofilm-responsive coat – it would be great to discuss what this coat material is composed of, and what component acts as the trigger that would release the phages specifically at biofilms. Similarly, what are the techniques used to engineer bacteriophages. Such topics may be explained in a couple of sentences, which when done at all the places in the review will make it even more interesting for the readers.

Dear Reviewer-2,

Thank you very much for your thoughtful and detailed comments on our manuscript titled “A Comprehensive Review on Phage Therapy and Phage-Based Drug Development.”.  We greatly appreciate your encouraging feedback and valuable suggestions, which have helped us improve the quality of the review. We have carefully addressed each of your points and made corresponding revisions to the manuscript. Below, we provide our responses to your comments and detail the changes we made.

  1. The review is very lengthy. This can be shortened by eliminating general/redundant statements, which can be found at several places in the review. For example, the lines 239-240, 414-416, 446-448, 553-554, etc. have almost the same meaning. Moreover, removing those statements at most places will not change the way the message is communicated.

Response:

We appreciate your suggestion regarding the length of the review. We have carefully reviewed the manuscript and eliminated redundant statements as you indicated, including the specific lines mentioned (239-240, 414-416, 446-448, 553-554), along with others throughout the text. These changes have resulted in a more concise and focused manuscript without compromising the clarity or overall message. This revision should enhance the readability and impact of the review for a broader audience.

  1. After the word-count is cut down significantly, it would be great if the authors can make an attempt to critically analyze/discuss the work they have cited. For example, in lines 427-429 the authors write that Zuo et al. developed a biofilm-responsive coat – it would be great to discuss what this coat material is composed of and what component acts as the trigger that would release the phages specifically at biofilms. Similarly, what are the techniques used to engineer bacteriophages. Such topics may be explained in a couple of sentences, which when done at all the places in the review will make it even more interesting for the readers.

Response:

Thank you for this insightful recommendation. We have revised the manuscript to include a more critical analysis and discussion of the cited works. Specifically, we expanded on the example you provided (lines 427-429), elaborating on the biofilm-responsive coat developed by Zuo et al. (2022). As outlined in revised manuscript, this study describes a biofilm-responsive encapsulated-phage coating that autonomously detects and responds to biofilm formation by releasing phages, which disrupt the biofilm matrix. This approach also highlights potential implications for modulating immune-signaling pathways when applied in biomedical contexts. Additionally, we have incorporated further detail on other key studies, including techniques used to engineer bacteriophages, such as CRISPR-Cas systems and directed evolution. These revisions should offer readers a deeper understanding and enhance the review’s critical depth..

We hope the revisions we have made address your concerns and improve the overall quality of the manuscript. We are confident that the revised version will be more engaging and informative for both young researchers and senior experts. Thank you again for your valuable feedback.

Reviewer 3 Report

Comments and Suggestions for Authors

This review paper covers an impressive scope, encompassing nearly everything related to phages. Although there are some areas where details are slightly overlooked, I believe the authors have provided a thorough exploration and comprehensive review of drug development, which is their primary focus. Additionally, the paper’s emphasis on multidrug resistance is highly motivating and likely to capture the readers' interest. Therefore, I would like to offer a few suggestions for the paper's improvement:

1. Please ensure that the source of each figure, such as Figure 1, and any specific symbols used, is clearly cited.

2. Section 3.5 on combination therapy is particularly intriguing. Could you present this section in a condensed, tabular format?

3. Given the broad scope of the paper, it might be beneficial to remove chapters that are not closely related to human health.

4. Overall, this is an extensive and excellent paper. Even the conclusion effectively supports the results presented. Truly an excellent paper.

Author Response

Reviewer-3

Comments and Suggestions for Authors

This review paper covers an impressive scope, encompassing nearly everything related to phages. Although there are some areas where details are slightly overlooked, I believe the authors have provided a thorough exploration and comprehensive review of drug development, which is their primary focus. Additionally, the paper’s emphasis on multidrug resistance is highly motivating and likely to capture the readers' interest. Therefore, I would like to offer a few suggestions for the paper's improvement:

  1. Please ensure that the source of each figure, such as Figure 1, and any specific symbols used, is clearly cited.
  2. Section 3.5 on combination therapy is particularly intriguing. Could you present this section in a condensed, tabular format?
  3. Given the broad scope of the paper, it might be beneficial to remove chapters that are not closely related to human health.
  4. Overall, this is an extensive and excellent paper. Even the conclusion effectively supports the results presented. Truly an excellent paper.

Dear Reviewer-3,

Thank you very much for your kind and encouraging feedback on our manuscript titled “A Comprehensive Review on Phage Therapy and Phage-Based Drug Development”. We greatly appreciate your positive evaluation and insightful suggestions for improvement. We have carefully addressed each of your points, and below, we provide detailed responses and an outline of the changes we have made to the manuscript.

  1. Please ensure that the source of each figure, such as Figure 1, and any specific symbols used, is clearly cited.

Response:

Thank you for this observation. We have reviewed all figures in the manuscript, including Figure 1, and have ensured that the sources and references are clearly cited. Any specific symbols used in the figures have also been properly defined and referenced where applicable. We believe this will enhance the clarity and attribution of the visual elements in our paper.

  1. Section 3.5 on combination therapy is particularly intriguing. Could you present this section in a condensed, tabular format?

Response:

We appreciate your suggestion regarding Section 3.5. We agree that presenting the information in a tabular format could make it easier for readers to digest. Therefore, we have condensed the key information from the section on combination therapy into a table, (Inserted a new Table as Table 4) which highlights the types of therapies, their mechanisms of action, and examples from the literature. This revision should improve the accessibility of the content while maintaining the section’s depth.

  1. Given the broad scope of the paper, it might be beneficial to remove chapters that are not closely related to human health.

Response:

We appreciate your comment regarding the scope of the review. After careful consideration, we have decided to streamline the manuscript by removing certain sections, e.g. the Section 9 was rewrite and degreased it’s volume about half, that are less directly related to human health. This helps maintain a more focused narrative aligned with the primary themes of phage therapy and drug development, particularly in the context of human medicine. We believe this will make the review more cohesive and appealing to readers with a specific interest in health-related applications of phages.

  1. Overall, this is an extensive and excellent paper. Even the conclusion effectively supports the results presented. Truly an excellent paper.

Response:

Thank you for your very kind words. We are delighted that you found the paper comprehensive and well-structured. Your feedback has been invaluable in enhancing the quality of the review, and we hope the revisions further improve its impact.

We sincerely hope that the changes we have made in response to your comments address your concerns and improve the overall quality of the manuscript. Thank you again for your thoughtful suggestions and positive feedback.

Round 2

Reviewer 1 Report

Comments and Suggestions for Authors

The authors has improved the manuscript.